# A stable microtubule bundle formed through an orchestrated multistep process controls quiescence exit

Damien Laporte[1]*, Aurelie Massoni-Laporte[1], Charles Lefranc[1], Jim Dompierre[1], David Mauboules[1], Emmanuel T Nsamba[2], Anne Royou[1], Lihi Gal[3], Maya Schuldiner[3], Mohan L Gupta[2], Isabelle Sagot[1]

[1]Univ. Bordeaux, CNRS, IBGC, UMR 5095, Bordeaux, France; [2]Genetics, Development, and Cell Biology, Iowa State University, Ames, United States; [3]Department of Molecular Genetics, Weizmann Institute of Science, Rehovot, Israel

*For correspondence:
laporte@ibgc.cnrs.fr

Competing interest: The authors declare that no competing interests exist.

**Abstract** Cells fine-tune microtubule assembly in both space and time to give rise to distinct edifices with specific cellular functions. In proliferating cells, microtubules are highly dynamics, and proliferation cessation often leads to their stabilization. One of the most stable microtubule structures identified to date is the nuclear bundle assembled in quiescent yeast. In this article, we characterize the original multistep process driving the assembly of this structure. This Aurora B-dependent mechanism follows a precise temporality that relies on the sequential actions of kinesin-14, kinesin-5, and involves both microtubule–kinetochore and kinetochore–kinetochore interactions. Upon quiescence exit, the microtubule bundle is disassembled via a cooperative process involving kinesin-8 and its full disassembly is required prior to cells re-entry into proliferation. Overall, our study provides the first description, at the molecular scale, of the entire life cycle of a stable microtubule structure in vivo and sheds light on its physiological function.

## eLife assessment

This work presents **important** insights regarding the mechanism underlying the assembly, maintenance, and disassembly of a very stable microtubule-based structure, termed quiescent-cell nuclear microtubule (Q-nMT) bundle, which is formed in quiescent yeast cells to ensure cell survival and viability. This insight will help elucidate how very stable microtubules can exist alongside very dynamic microtubules, which is still poorly understood. While the experimental support is overall **solid**, additional analyses using state-of-the-art methodology would further strengthen some of the claims.

## Introduction

Microtubules (MTs) are hollow cylindrical polymers assembled by the non-covalent interaction of α- and β-tubulin heterodimers. MTs are generally nucleated at MT organizing centers (MTOCs) by a ɣ-tubulin complex (ɣ-TuC) that acts as a template and stabilizes the so-called MT 'minus-end' (*Liu et al., 2021*; *Roostalu and Surrey, 2017*; *Sanchez and Feldman, 2017*; *Thawani and Petry, 2021*). At the opposite end, the 'plus-end' (+end), MTs elongate by the addition of GTP tubulin. The β-tubulin bound GTP is hydrolyzed and stable but transient GDP + Pi intermediates are generated in the region immediately behind the growing +end. Subsequent Pi release favors MT depolymerization, which can be rescued by de novo GTP-tubulin addition (*Cleary and Hancock, 2021*; *Gudimchuk and McIntosh, 2021*). Thus, MTs alternate between periods of growth and shrinkage, a behavior termed

dynamic instability (*Mitchison and Kirschner, 1984*). In vivo, a profusion of MT-associated proteins (MAPs) regulate MT length and dynamics (*Bodakuntla et al., 2019*; *Goodson and Jonasson, 2018*). In addition, MTs are often assembled from multiple tubulin variants or isotypes, and are modified by a cohort of post-translational modifications that modulate MT dynamics directly or by influencing the recruitment of MAPs (*Janke and Magiera, 2020*; *Roll-Mecak, 2020*). Other MAPs organize MTs into high-ordered assemblies, either by cross-linking MTs or by connecting them with cellular structures such as membranes, chromosomes, or actin cytoskeleton components (*Bodakuntla et al., 2019*; *Meiring et al., 2020*). Cells fine-tune these mechanisms in both space and time to give rise to distinct MT edifices with specific functions.

In proliferating cells, MT dynamics are crucial for their functions. It allows exploration of the cell volume in search of structures to 'capture', such as centromeres during mitosis, or to exert pushing or pulling forces required for various cellular processes such as cell migration or morphogenesis (*Heald and Khodjakov, 2015*; *Kirschner and Mitchison, 1986*). The MT cytoskeleton can undergo extensive rearrangements and assemble more stable MT structures, especially when cells change fate (*Meiring et al., 2020*; *Röper, 2020*). For example, in terminally differentiated cells, such as epithelial cells, cardiomyocytes, or neurons, stable MT networks make up the majority and ensure critical cellular functions such as cell shape maintenance or long-distance intracellular transport (*Baas et al., 2016*; *Muroyama and Lechler, 2017*). Defects in MT stabilization are at the origin of many human pathologies, including neurodegenerative diseases and ciliopathies (*Anvarian et al., 2019*; *Wheway et al., 2018*). Effectors responsible for MT stabilization have been actively searched for since the 1980s. In mammals, the involvement of specific MAPs such as Tau, MAP2, MAP6, and PRC1, or tubulin post-translational modifications, has been extensively studied. While their contributions to MT stabilization are central and undisputed, their sole actions do not fully explain the observed levels of MT stability in many types of non-dividing cells (*Hahn et al., 2019*).

For years, yeast species have been instrumental in unraveling the mechanisms that regulate MT dynamics in eukaryotes. While proliferating yeast cells exhibit a dynamic MT network (*Winey and Bloom, 2012*), proliferation cessation goes with the formation of dramatically stable MT structures in both *Saccharomyces cerevisiae* and *Schizosaccharomyces pombe* (*Laporte and Sagot, 2014*; *Laporte et al., 2013*; *Laporte et al., 2015*). Upon quiescence establishment, *S. cerevisiae* cells assemble a bundle composed of stable parallel MTs, hereafter referred to as the Q-nMT bundle, for *Q*uiescent-cell *n*uclear *M*icro*t*ubule bundle. This structure originates from the nuclear side of the spindle pole body (SPB), the yeast equivalent of the centrosome. It spans the entire nucleus and relocalizes kinetochores and centromeres, which remain attached to MT +ends (*Laporte and Sagot, 2014*). Because the MTs embedded in the Q-nMT bundle are not all of the same length, the structure has a typical arrow shape. When cells exit quiescence, the Q-nMT bundle depolymerizes and, by pulling the attached centromeres back to the SPB, allows the recovery of the typical Rabl-like configuration of chromosomes found in G1 yeast cells (*Jin et al., 1998*). The molecular mechanisms underlying the formation of this peculiar stable MT structure and its physiological function(s) are not understood. However, cells defective in Q-nMT bundle formation have a compromised survival in quiescence and a drastically reduced fitness upon cell cycle re-entry (*Laporte and Sagot, 2014*; *Laporte et al., 2013*; *Laporte et al., 2015*).

Here, we show that Q-nMT bundle formation is a multistep process that follows a precise temporal sequence. The first step relies on Aurora B/Ipl1 activity and requires the kinesin-14 Kar3, its regulator Cik1, and the EB1 homolog Bim1. It leads to the formation of a short ($\approx$ 1 µm) and stable bundle that resembles a half mitotic spindle. Importantly, in this first step, MT polymerization and stabilization are coupled. In a second step, additional MTs polymerize from the SPB, elongate, and are zipped to pre-existing MTs in a Cin8/kinesin-5-dependent manner. Complete stabilization of the Q-nMT bundle is achieved in a third and final step that requires the kinesin-5 Kip1. Our observations further indicate that Q-nMT bundle assembly requires both MT–kinetochore attachment and inter-kinetochore interactions. Finally, we show that, upon exit from quiescence, the Q-nMT bundle disassembles from its +ends, each MT depolymerizing in coordination with its neighbors, via the action of the depolymerase Kip3, a member of the kinesin-8 family. Importantly, we show that the complete disassembly of the Q-nMT bundle is required for cells to re-enter the proliferation cycle. Overall, this study describes the entire life cycle of an MT structure, from the molecular mechanisms involved in its formation and stabilization to its disassembly, and further suggests that this atypical

quiescent-specific structure could act as a regulator, or a 'control point' for cell cycle resumption upon exit from quiescence.

## Results

### The Q-nMT bundle formation is a three-step process

We have previously shown that the Q-nMT bundle observed in quiescent cells is a stable nuclear structure (*Laporte et al., 2013*). To investigate whether the stabilization of the Q-nMT bundle was concomitant with its assembly or whether it assembled first as a dynamic structure and then gets stabilized, we quantified the Q-nMT bundle length (*Figure 1A*) and thickness (*Figure 1B*) upon quiescence establishment following carbon source exhaustion. We found that the Q-nMT bundle assembly process could be divided into three sequential phases. In an initial phase (phase I), MTs elongated from the SPB to reach ≈ 0.8 µm. The number of these MTs, referred to as phase I-MTs, was approximately the same as in a mitotic spindle (see inset of *Figure 1B* and *Figure 1—figure supplement 1A*). Importantly, phase I-MTs were resistant to nocodazole (Noc), a MT poison that causes dynamic MT depolymerization, indicating that phase I-MT stabilization was concomitant with their polymerization (*Figure 1A and B*). In a second phase, beginning ≈ 10 hr after glucose exhaustion, additional MTs emerged from the SPB (phase II-MTs) and elongated along phase I-MTs, nearly doubling the thickness of the phase I MT bundle (*Figure 1B*, *Figure 1—figure supplement 1A*). At this stage, the tip of the newly elongated MTs was unstable (*Figure 1A and B*). Complete stabilization of the Q-nMT bundle was achieved ≈ 48 hr after glucose exhaustion (phase III). Indeed, after this step, a Noc treatment did not affect either the length or the thickness of the Q-nMT bundle (*Figure 1A and B*, *Figure 1—figure supplement 1B*). Plotting Q-nMT bundle width as a function of length for each individual cell before and after Noc treatment further revealed the three successive phases of the Q-nMT bundle formation (*Figure 1C*). Of note, Q-nMT bundles were also observed both in diploid cells upon glucose exhaustion (*Figure 1—figure supplement 1C*) and in haploid cells transferred to water (*Figure 1—figure supplement 1D*).

To confirm the above observations, first, we followed Nuf2-GFP, a protein that localizes to the MT +end. As expected, movies showed that upon Q-nMT bundle formation initiation, the Nuf2-GFP signal relocated from the SPB to the distal part of elongating Q-nMT bundles (*Figure 1—figure supplement 1E*). In phase II, when cells were treated with Noc, the Nuf2-GFP signal followed depolymerizing MTs (*Figure 1D*, left panel), testifying for the instability of phase II-MTs. In contrast, the Nuf2-GFP signal remained immobile upon Noc treatment in phase III as cells have fully stabilized Q-nMT bundles (*Figure 1D*, right panel). Second, we measured the γ-tubulin (Tub4) signal at the SPB and found that Tub4 started to accumulate at the SPB at the beginning of phase I to reach a plateau at the end of phase II (*Figure 1E*, *Figure 1—figure supplement 1F*). Interestingly, the amount of Tub4 at the SPB was proportional to the thickness of the Q-nMT bundle (*Figure 1—figure supplement 1G*). Finally, in order to confirm the two waves of MT elongation, we developed a strain expressing mTQZ-Tub1 from the endogenous TUB1 promoter and mRuby-TUB1 under the ADH2 promoter, a promoter that is de-repressed after glucose exhaustion, as confirmed by western blot (*Figure 1—figure supplement 1H*). We observed that mRuby-TUB1 was incorporated into the Q-nMT bundle only after phase I (*Figure 1F*, *Figure 1—figure supplement 1H*), confirming the existence of a second wave of MT elongation during phase II. All the above experiments led us to propose the model shown in *Figure 1G*.

### Q-nMT bundle formation is an essential time-regulated process

Increasing cytoplasmic viscosity has been shown to dampen MT dynamics (*Molines et al., 2022*). Upon quiescence establishment, yeast cells undergo a transition from a fluid to a solid-like phase (*Munder et al., 2016*; *Joyner et al., 2016*) and an acidification (*Jacquel et al., 2021*), which could contribute to the formation of the Q-nMT bundle. Several lines of evidence suggest that these changes in physicochemical properties are not involved in Q-nMT formation. First, 4-day-old quiescent cells can simultaneously display both dynamic cytoplasmic MTs (cMTs) and a stable Q-nMT bundle (*Figure 1—figure supplement 1I* and *Laporte et al., 2013*). Second, when we mimicked a fluid to solid-like phase transition in proliferating cells by artificially lowering the pH, no stable MTs were observed, whereas F-actin aggregation was induced (*Figure 1—figure supplement 1J*) as expected (*Peters et al., 2013*). Third,

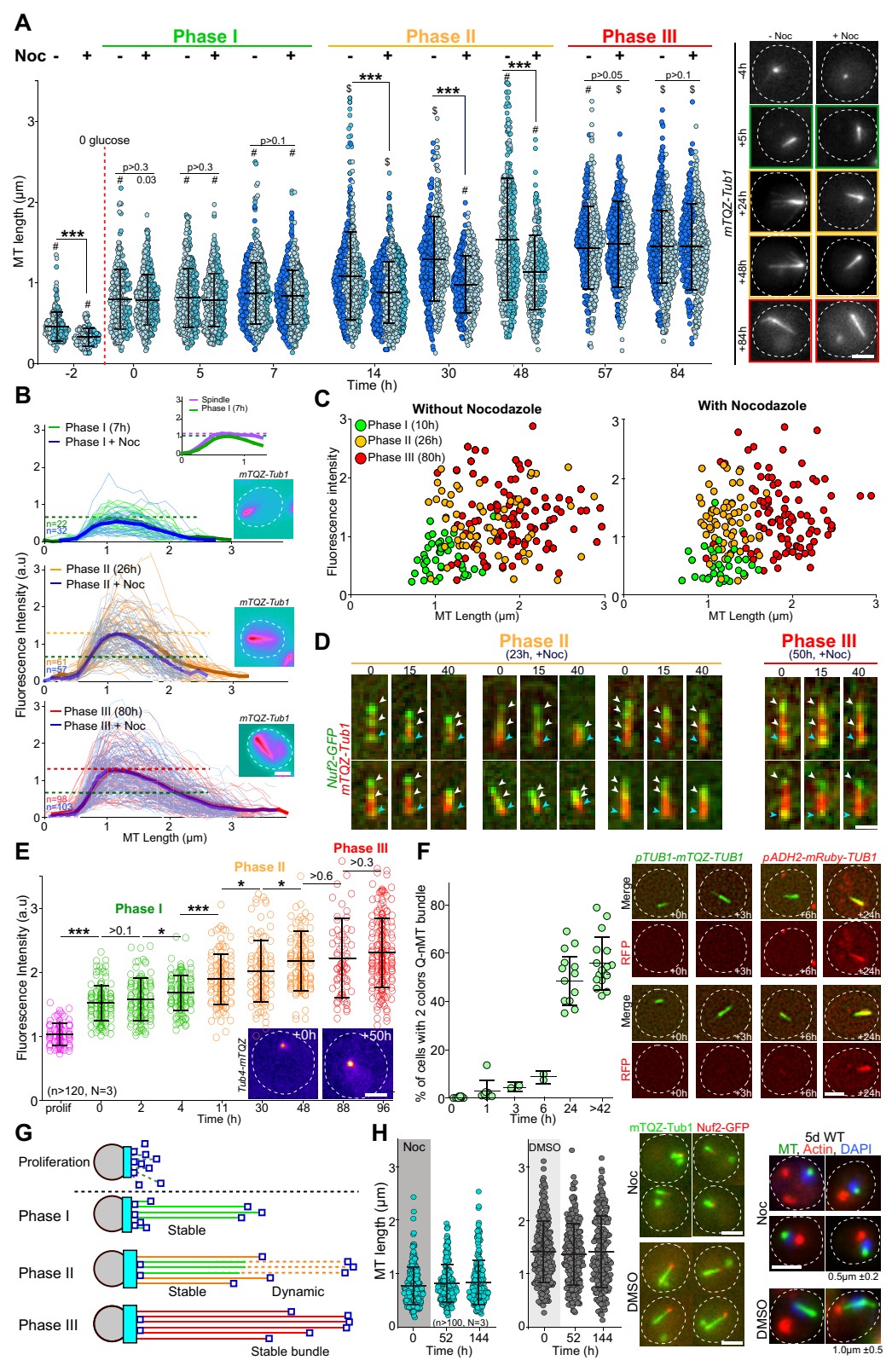

**Figure 1.** The formation of the quiescent-cell nuclear microtubule (Q-nMT) bundle is a three-step process. (**A**) Nuclear MT length in WT cells expressing mTQZ-Tub1, before (**-**) or after (**+**) a 15 min Noc treatment (30 µg/ml) upon entry into quiescence. Each circle corresponds to the length of an individual MT structure. Three independent experiments are shown (pale blue, cyan and dark blue, n > 160 for each point in each experiment).

*Figure 1 continued on next page*

*Figure 1 continued*

The mean and SD are shown. Student's test or ANOVA (sample >2) were used to compare inter-replicates. #p-value>0.05, $p-value<0.05. A Student's test (*t*-test with two independent samples) was used to compare the results obtained with or without Noc, the indicated p-values being the highest measured among experiments. ***p-value<$1.10^{-5}$. Images of representative cells are shown. Bar is 2 μm. (**B**) MT fluorescence intensity as a proxy of MT structure width in WT cells expressing mTQZ-Tub1. Mean intensity measurement for half pre-anaphase mitotic spindles (purple), phase I (green), phase II (orange), or phase III (red) Q-nMT bundle. A line scan along the MT structure for individual cells is shown as thin lines, the mean as a bold line (n > 60/phase), all the lines being aligned at 0.5 μm before the fluorescence intensity increase onset on the spindle pole body (SPB) side. The blue lines are results obtained after a 15 min Noc treatment (30 μg/ml). In each graph, horizontal dashed line indicates the mean intensity. Images in pseudo-colors of a representative cell for each phase are shown. Bar is 2 μm. (**C**) MT bundle length as a function of MT bundle width for individual cells in each phase before and after a 15 min Noc treatment (30 μg/ml) in WT cells expressing mTQZ-Tub1. Each circle represents an individual MT structure. (**D**) WT cells expressing mTQZ-Tub1 (red) and Nuf2-GFP (green) in phase II (23 hr) or phase III (50 hr) were deposited on an agarose pad containing 30 μg/ml Noc and imaged. Blue arrowheads: SPB; white arrowheads: Nuf2-GFP clusters. Time is in min after deposition on the pad. Bar is 1 μm. (**E**) Tub4-mTQZ fluorescence intensity measured at the SPB upon entry into quiescence. Each circle represents an individual cell. The mean and SD are shown; *t*-tests were used to compare independent samples (N = 3, n > 150), *0.05>p-value>$1.10^{-3}$, ***p-value<$1.10^{-5}$. Images in pseudo-colors of representative cells are shown. Bar is 2 μm. (**F**) WT cells expressing mTQZ-Tub1 under the *TUB1* promoter and mRuby-Tub1 under the *ADH2* promoter. The percentage of cells harboring both mTQZ and mRuby fluorescence along the Q-nMT bundle is shown; each circle being the percentage for an independent experiment, with n > 200 cells counted for each experiment. The mean and SD are shown. Images of representative cells at the indicated time after glucose exhaustion are shown. Bar is 2 μm. (**G**) Schematic of the Q-nMT bundle formation. During phase I, stable MTs (phase I-MT, green) elongate from the SPB (gray). During phase II, the amount of Tub4 (cyan) increases at the SPB. In the meantime, new MTs (phase II-MT, orange) elongated from the SPB and are stabilized along the phase I-MTs, yet their +ends remain dynamic (dashed lines). After phase III, all MTs are stabilized (red). Nuf2 is schematized as dark blue squares. (**H**) Upon glucose exhaustion, WT cells expressing mTQZ-Tub1 (green) and Nuf2-GFP (red) were pulsed treated with 30 μg/ml Noc (blue) or DMSO (gray) for 24 hr. Noc or DMSO were then chased using carbon-exhausted medium and cells were imaged. Each circle corresponds to MT structure length in an individual cell. The mean and SD are shown (N = 3, n > 100). Images of representative cells 2 d after the chase and representative cells 5 d after the chase are shown. For the right panels, tubulin (green) was detected by immunofluorescence, actin (red) by phalloidin, and DNA (blue) with DAPI. The mean Q-nMT bundle length (± SD) in the population is indicated.

The online version of this article includes the following source data and figure supplement(s) for figure 1:

**Source data 1.** Excel file containing MT length measurement upon entry into quiescence, before or after Noc treatment.

**Source data 2.** Excel file containing MT intensity measurement upon entry into quiescence, before or after Noc treatment.

**Source data 3.** Excel file containing individual MT bundle length as a function of its width.

**Source data 4.** Excel file containing fluorescence intensities measurement of Tub4 at the SPB during quiescence entry.

**Source data 5.** Excel file containing MT length measurement in individual cells, after DMSO or Noc pulse chase.

**Figure supplement 1.** The three steps of quiescent-cell nuclear microtubule (Q-nMT) bundle formation.

**Figure supplement 1—source data 1.** Excel file containing nuclear MT structure intensity measurements in WT haploid or diploid cells.

**Figure supplement 1—source data 2.** Excel file containing the fluorescence intensity measurements of Tub4 and Tub1 in individual cells.

**Figure supplement 1—source data 3.** Excel file containing WT cell viability after the indicated treatment.

**Figure supplement 1—source data 4.** Uncropped western blot for *Figure 1—figure supplement 1H*.

**Figure supplement 1—source data 5.** Raw western blot for *Figure 1—figure supplement 1H* – Ade13.

**Figure supplement 1—source data 6.** Raw western blot for *Figure 1—figure supplement 1H* – GFP.

**Figure supplement 1—source data 7.** Raw western blot for *Figure 1—figure supplement 1H* – RFP.

2-day-old quiescent cells exhibiting a phase III Q-nMT bundle were able to grow de novo dynamic cMTs after a pulse-chase Noc treatment (*Figure 1—figure supplement 1K*).

Importantly, if Q-nMT bundle formation was solely dependent on physicochemical changes, this structure should be able to assemble in late quiescence. By treating cells with Noc upon glucose exhaustion, we were able to prevent the formation of the Q-nMT bundle in early quiescence (*Figure 1H*). Strikingly, when the drug was washed out, no Q-nMT bundle assembled, even 144 hr after Noc removal (*Figure 1H*). These cells had entered quiescence because they had assembled actin bodies (*Figure 1H*, right panel), another quiescent cell-specific structure (*Sagot et al., 2006*). This experiment strongly suggests that Q-nMT formation is a process induced by a transient signal emitted upon glucose exhaustion.

We have previously shown that mutants unable to assemble Q-nMT bundles have reduced viability in quiescence and a reduced ability to form colonies upon exit from quiescence (*Laporte et al., 2013*). To establish a direct link between the absence of the Q-nMT bundle and the above phenotypes, we took advantage of our ability to conditionally prevent Q-nMT bundle formation in WT cells using Noc treatment at the onset of glucose exhaustion. As shown in *Figure 1—figure supplement 1L*, in the absence of the Q-nMT bundle, WT cells lose viability in quiescence and survivors have a reduced ability to generate a progeny upon exit from quiescence. In contrast, a similar Noc treatment of cells that have already assembled a stable Q-nMT bundle (5-day-old cells) did not affect either the viability in quiescence or quiescence exit efficiency, demonstrating that Noc did not have a toxic effect per se. Taken together, these experiments confirm our initial findings that the Q-nMT bundle is important for both cell survival during chronological aging and quiescence exit fitness.

## Steady-state tubulin levels and Tub3 isotype are critical for Q-nMT bundle formation

We then focused on deciphering the molecular mechanisms involved in Q-nMT bundle formation. Yeast cells express two α-tubulin isotypes, Tub1 and Tub3. In proliferating cells, Tub1 constitutes the majority (*Aiken et al., 2019*; *Nsamba et al., 2021*; *Schatz et al., 1986*). In 4-day-old cells, the two α-tubulin isotypes were found embedded in the Q-nMT bundle. This was observed using two different pairs of fluorescent proteins in order to avoid any influence of the FP variants and using both widefield and expansion microscopy (*Figure 2A*, *Figure 2—figure supplement 1A*). Since Tub3 stabilizes MTs in vitro (*Bode et al., 2003*), it may be key for Q-nMT bundle formation. To test this hypothesis, we first analyzed the phenotype of *tub3Δ* cells. In phase II, *tub3Δ* cells displayed short MT bundles that were sensitive to Noc. In phase III, no MT bundles were detected (*Figure 2B*, *Figure 2—figure supplement 1C*). Thus, in this mutant, the MT bundles assembled during phase I were not stable and eventually collapsed. In *tub3Δ* cells, the steady-state amount of α-tubulin is significantly reduced (*Figure 2—figure supplement 1B* and *Nsamba et al., 2021*). Thus, the amount of α-tubulin, but not Tub3 itself, may be important for Q-nMT bundle formation. Indeed, mutants defective in α- and β-tubulin folding, such as the prefolding complex mutants *pac10Δ*, *gim3Δ*, or *yke2Δ*, or in tubulin heterodimer formation such as *pac2Δ*, or in β-tubulin folding, such as *cin2Δ* or *cin4Δ*, all of which have a reduced amount of tubulin, were unable to assemble a Q-nMT bundle (*Figure 2—figure supplement 1D and E*).

To identify the role of Tub3 per se, we took advantage of the 'Tub1-only' mutant, in which the *TUB3* gene was replaced by the *TUB1* gene at the *TUB3* locus. This mutant expresses only Tub1 and has an α-tubulin level similar to WT (*Figure 2—figure supplement 1B* and *Nsamba et al., 2021*). We found that 'Tub1-only' cells have unstable Q-nMT bundles in phase II, and shorter, yet stable, Q-nMT bundles in phase III (*Figure 2C*, *Figure 2—figure supplement 1C*). Moreover, 'Tub1-only' Q-nMT bundles were thinner than WT Q-nMT bundles (*Figure 2D*). Taken together, our data indicate that Tub3 is involved in MT elongation in phase II since in the absence of this isotype phase I MT bundles could assemble and MT bundles could get stabilized in phase III, but the second wave of MT nucleation and elongation was impaired as testified by thinner Q-nMT bundles in both phases II and III.

## Kinetochores are critical for phase I but dispensable for the maintenance of Q-nMT bundles

Kinetochore components are enriched at the Q-nMT bundle +end (*Laporte et al., 2013* and *Figure 3—figure supplement 1A*). We therefore tested whether kinetochore–MT attachment might play a role in Q-nMT bundle formation. First, we disrupted kinetochore–MT attachment upon glucose

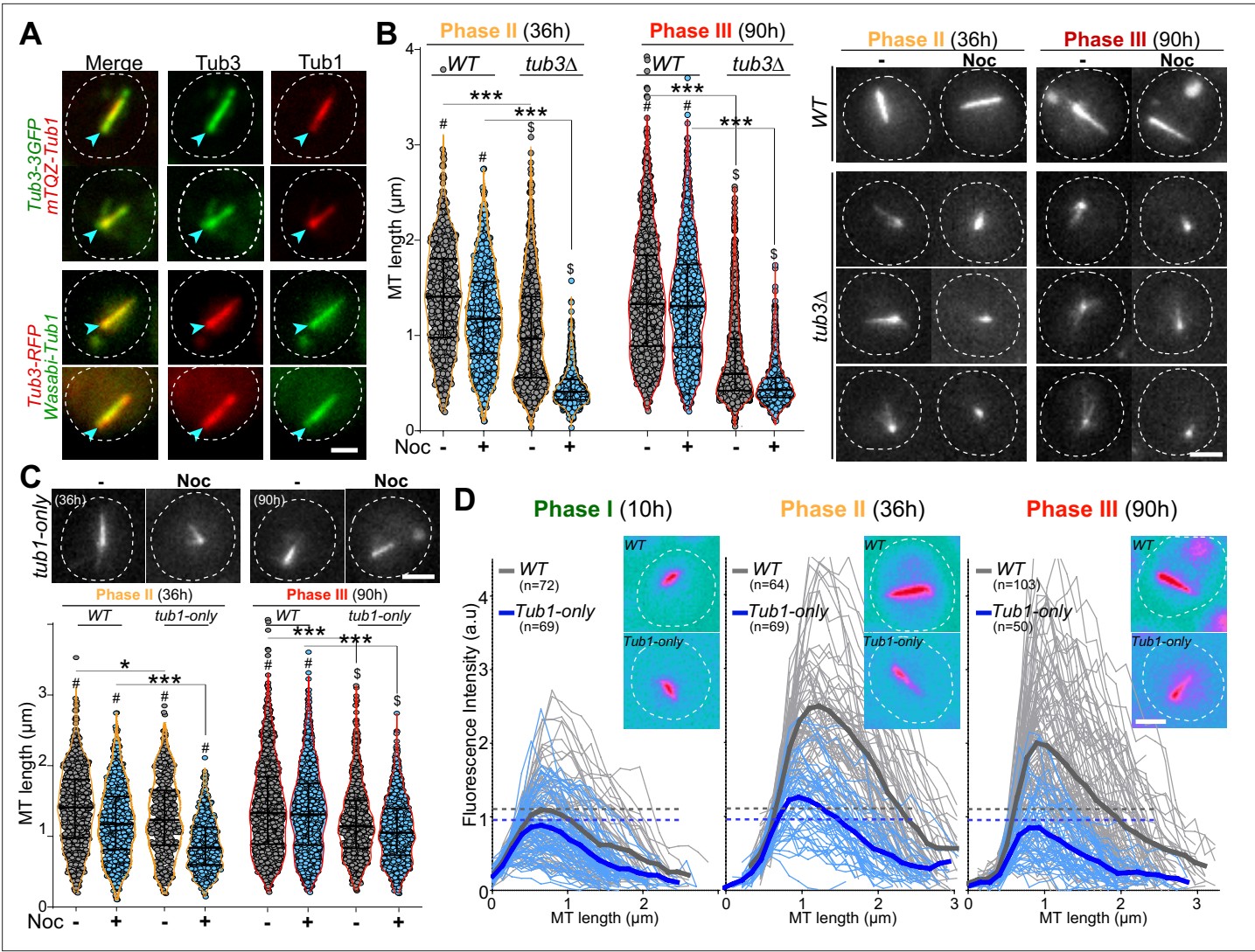

**Figure 2.** Quiescent-cell nuclear microtubule (Q-nMT) bundle formation is influenced by the α-tubulin amount and isotype. (**A**) WT cells (4 d) expressing either Tub3-3GFP (green) and mTQZ-Tub1 (red, top panel) or Tub3-RFP (red) and mWasabi-Tub1 (green, bottom panel). Blue arrowheads point to spindle pole body (SPB). Bar is 2 μm. (**B**) Nuclear MT length in WT and *tub3Δ* cells expressing mTQZ-Tub1, 36 hr (phase II, orange) and 90 hr (phase III, red) after glucose exhaustion, treated 15 min (blue) or not (gray) with 30 μg/ml Noc. Each circle corresponds to the length of an individual MT structure, mean and SD are shown. ANOVA was used to compare inter-replicates (n > 200, N = 5); #p-value>0.05, $p-value<0.05. A Student's test (*t*-test with two independent samples) was used to compare (+) or (-) Noc data. The indicated p-values are the highest calculated among the five experiments; ***p-value<$1.10^{-5}$. Images of representative cells are shown. Bar is 2 μm. (**C**) Nuclear MT length in WT and *Tub1-only* cells expressing mTQZ-Tub1, 36 hr (phase II, orange) and 90 hr (phase III, red) after glucose exhaustion, treated 15 min or not with 30 μg/ml Noc. Statistical representations are as in (**B**). Images of representative cells are shown, bar is 2 μm. (**D**) Fluorescence intensity along the Q-nMT bundle in WT (gray) and *Tub1-only* (blue) cells expressing mTQZ-Tub1 grown for 10 hr (phase I), 36 hr (phase II), and 90 hr (phase III). Individual fluorescent intensity is shown as thin line, the mean as a bold line, all the lines being aligned at 0,5 μm before the fluorescence intensity increase onset on the SPB side. Dashed lines indicate the maximal mean fluorescence intensity. Images in pseudo-color of representative cells are shown. Bar is 2 μm.

The online version of this article includes the following source data and figure supplement(s) for figure 2:

**Source data 1.** Excel file containing nuclear MTs length measurements in different strains backgrounds.

**Source data 2.** Excel file containing Q-nMT bundles fluorescence intensity measurement in different strain backgrounds.

**Figure supplement 1.** The impact of the α-tubulin level on quiescent-cell nuclear microtubule (Q-nMT) bundle assembly.

**Figure supplement 1—source data 1.** Uncropped western blot for *Figure 2—figure supplement 1B*.

**Figure supplement 1—source data 2.** Uncropped western blot for *Figure 2—figure supplement 1D*.

**Figure supplement 1—source data 3.** Uncropped western blot for *Figure 2—figure supplement 1E*.

*Figure 2 continued on next page*

*Figure 2 continued*

**Figure supplement 1—source data 4.** Raw western blot for *Figure 2—figure supplement 1B*.

**Figure supplement 1—source data 5.** Raw western blot for *Figure 2—figure supplement 1D* – loading control.

**Figure supplement 1—source data 6.** Raw western blot for *Figure 2—figure supplement 1D*.

**Figure supplement 1—source data 7.** Raw western blot for *Figure 2—figure supplement 1E* – loading control.

**Figure supplement 1—source data 8.** Raw western blot for *Figure 2—figure supplement 1E*.

exhaustion using the well-established *ndc80-1* allele (*Cheeseman et al., 2006*; *DeLuca et al., 2018*; *Wigge et al., 1998*). As shown in *Figure 3A*, only very short MT structures were detected in *ndc80-1* cells transferred to 37°C at the onset of quiescence entry, even after 4 d at non-permissive temperature (for control, see *Figure 3—figure supplement 1B*). Second, we focused on the chromosome passenger complex (CPC: Bir1/Survivin, Sli15/INCENP, Nbl1/Borealin, and Ipl1/Aurora B), a complex known to regulate kinetochore–MT attachment dynamics (*Cairo and Lacefield, 2020*). We found that inactivation of *IPL1* upon entry into quiescence using the thermosensitive allele *ipl1-1* prevented Q-nMT bundle formation (*Figure 3—figure supplement 1C*). The absence of Q-nMT bundles in cells harboring the NA-PP1 kinase-sensitive allele *ipl1-5as* (*Nerusheva et al., 2014*) indicated that the Ipl1 kinase activity was required for this process (*Figure 3B*, *Figure 3—figure supplement 1D and E* for controls). While we could not detect Ipl1 and Nbl1 in quiescent cells, we found that Sli15-GFP and Bir1-GFP localized along the Q-nMT bundle with an enrichment at the Q-nMT bundle +end (*Figure 3C*). In 4-day-old *bir1Δ* cells, Q-nMT bundle were shorter and not fully stabilized (*Figure 3D*). In addition, inactivation of *SLI15* upon glucose exhaustion using the thermosensitive allele *sli15-3* (*Kim et al., 1999*) drastically impaired Q-nMT bundle formation (*Figure 3—figure supplement 1F*). Finally, we examined the involvement of Bim1. Bim1 is a MT +end binding protein that plays a role in kinetochore MT-end on attachment (*Dudziak et al., 2021*; *Thomas et al., 2016*) and localizes along the entire Q-nMT bundle (*Laporte et al., 2013*). In fact, we found that that the amount of Bim1 correlated with tubulin incorporation during Q-nMT bundle formation (*Figure 3—figure supplement 1G*). *BIM1* deletion alone has no effect on Q-nMT bundle formation even in the presence of Noc (*Laporte et al., 2013* and *Figure 3D*). However, *bim1Δ bir1Δ* cells barely assembled MT structures, and these structures were sensitive to Noc (*Figure 3D*). This demonstrates that in the absence of Bim1, Bir1 is critical for Q-nMT bundle stabilization in phase I. Of note, we found that the viability of *bim1Δ bir1Δ* cells in quiescence was drastically reduced (*Figure 3—figure supplement 1H*). Together, these experiments show that kinetochore–MT attachments are critical for the initiation of Q-nMT bundle formation upon entry into quiescence.

We then hypothesized that inter-kinetochore interactions could stabilize parallel MTs embedded in the Q-nMT bundle. To test this idea, we analyzed MT organization in *slk19Δ* cells, which are defective in kinetochore clustering (*Richmond et al., 2013*). In quiescence, Slk19-GFP was enriched at both ends of the Q-nMT bundles (*Figure 3E*). *slk19Δ* cells displayed shorter and thinner MT structures that were sensitive to Noc (*Figure 3F*). Since Bim1 has been implicated in kinetochore–kinetochore interaction, we combined *slk19Δ* with *bim1Δ*. MT structures detected in *slk19Δ bim1Δ* cells barely reached ≈1 µm and disappeared upon Noc treatment. In addition, in *slk19Δ bim1Δ* cells, kinetochores localized as a rosette around the SPB (*Figure 3F*) and in *slk19Δ bim1Δ* cell viability was severely impaired (*Figure 3—figure supplement 1H*). This demonstrates that in the absence of Bim1, Slk19 is required for MT bundling and stabilization in phase I. We then focused on monopolin (Mam1, Lrs4, Hrr25, and Csm1), a complex that is known to cross-link kinetochores (*Corbett et al., 2010*). These proteins were found to be expressed in quiescent cells (*Figure 3—figure supplement 1I*), and when we analyzed the phenotype of *mam1Δ, lrs4Δ*, and *csm1Δ* cells in quiescence, most of them displayed short MT structures often arranged in a star-like array (*Figure 3G and H*, *Figure 3—figure supplement 1J*). As for *bir1Δ* and *slk19Δ*, the combination of *bim1Δ* with monopolin deletion worsened MT structure length and stability (*Figure 3H*), as well as cell viability in quiescence (*Figure 3—figure supplement 1H*). Furthermore, cells deleted for *SPO13*, a gene encoding a protein involved in monopolin recruitment to kinetochores (*Lee et al., 2004*), showed similar phenotypes (*Figure 3G and H*, *Figure 3—figure supplement 1H*). Collectively, these experiments demonstrate that inter-kinetochore interactions are critical for Q-nMT bundle assembly.

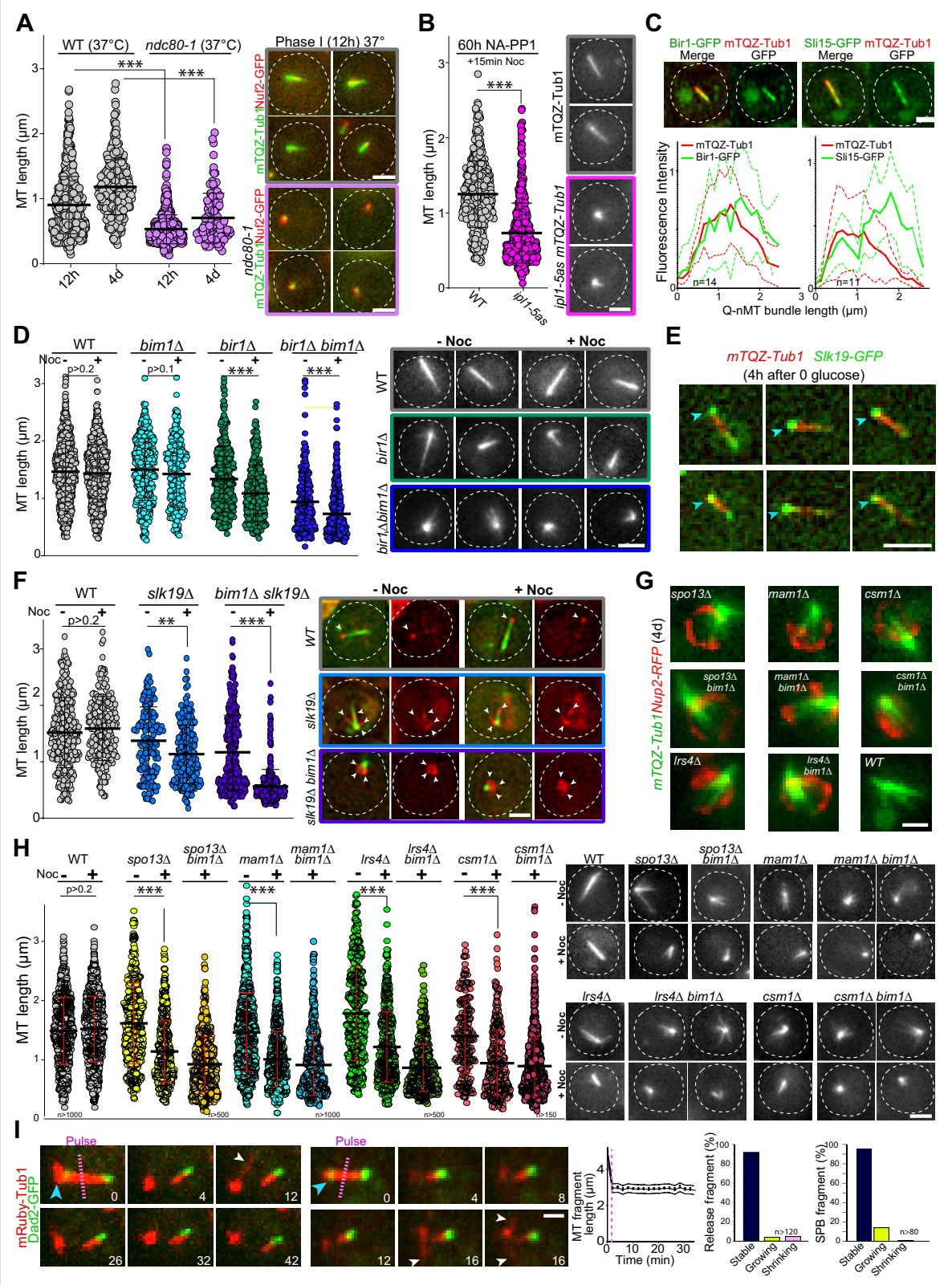

**Figure 3.** Kinetochore–kinetochore interactions are required for quiescent-cell nuclear microtubule (Q-nMT) bundle formation. (**A**) Nuclear MT length distribution in WT (gray) and *ndc80-1* (violet) cells expressing mTQZ-Tub1 (green) and Nuf2-GFP (red), transferred to 37°C upon glucose exhaustion and maintained at 37°C for the indicated time. Cells were imaged after a 20 min Noc treatment (30 µg/ml). See *Figure 3—figure supplement 1B* for the control without Noc. Each circle corresponds to the length of an individual MT structure, mean and SD are shown. A Student's test (*t*-test with two

*Figure 3 continued on next page*

*Figure 3 continued*

independent samples) was used to compare (+) or (-) Noc samples from the same experiment (n > 250, N = 2). The indicated p-values are the highest p-values calculated among the two repeated experiments. ***p-value<$1.10^{-5}$. Images of representative cells incubated 12 hr at 37°C are shown. Bar is 2 µm. (**B**) WT (gray) and *ipl1-5as* (pink) cells expressing mTQZ-Tub1 were treated for 60 hr with 50 µM NA-PP1 after glucose exhaustion, and imaged after a 15 min Noc treatment (30 µg/ml), see *Figure 3—figure supplement 1D* for control without Noc. Same statistical representation as in (**A**); N = 2, n > 200. Images of representative cells are shown. Bar is 2 µm. (**C**) WT cells (2 d) expressing mTQZ-Tub1 (red) and Bir1-GFP or Sli15-GFP (green) were imaged. Graphs show Bir1-GFP or Sli15-GFP fluorescence intensity along normalized Q-nMT bundles (see 'Materials and methods' section; plain and dash lines: mean and SD, respectively). Images of representative cells are shown. Bar is 2 µm. (**D**) Nuclear MT length distribution in cells of the indicated genotype (4 d) expressing mTQZ-Tub1 treated or not with Noc (30 µg/ml). Same statistical representation as in (**A**); N = 2, n > 200. Images of representative cells are shown. Bar is 2 µm. (**E**) WT cells expressing mTQZ-Tub1 (red) and Slk19-GFP (green) 4 hr after glucose exhaustion. Blue arrowhead: spindle pole body (SPB). Bar is 2 µm. (**F**) Nuclear MT length distribution in cells of the indicated genotype (4 d) expressing mTQZ-Tub1 (green) and Nuf2-GFP (red) imaged before or after Noc treatment (30 µg/ml). Same statistical representation as in (**A**); N = 2, n > 200. Images of representative cells are shown. White arrowheads point to Nuf2-GFP dots. Bar is 2 µm. (**G**) Cells of the indicated genotype (4 d) expressing mTQZ-Tub1 (green) and Nup2-RFP (red). Bar is 1 µm. (**H**) Nuclear MT length distribution in cells of the indicated genotype (4 d) expressing mTQZ-Tub1 treated or not with Noc (30 µg/ml). Same statistical representation as in (**A**); N ≥ 2, n > 200. Images of representative cells are shown. Bar is 2 µm. (**I**) Length variation of nuclear MT bundle fragments after laser ablation (pink dash line) in cells expressing mRuby-TUB1 (red) and Dad2-GFP (green). Time is in min. Images of representative cells are shown, blue arrowhead: SPB; white arrowhead: cytoplasmic MT (cMT). Graph indicates the variation of length in released fragments (n > 120) and histograms show the percentage of released or SPB attached fragments that are either stable, or that shorten or grow within a 30 min period after the laser-induced breakage. Bar is 1 µm.

The online version of this article includes the following source data and figure supplement(s) for figure 3:

**Source data 1.** Excel file containing nuclear MT length measurements in indicated strain backgrounds.

**Source data 2.** Excel file containing nuclear MT length measurements in WT and ipl1-5as cells.

**Source data 3.** Excel file containing measurement of the fluorescence intensity along Q-nMT bundles in Bir1-GFP or Sli15-GFP expressing cells.

**Source data 4.** Excel file containing the nuclear MT length distribution in indicated strain background.

**Source data 5.** Excel file containing the nuclear MT length distribution in indicated strains background.

**Source data 6.** Excel file containing the nuclear MT length distribution in indicated strains background.

**Figure supplement 1.** Kinetochore–microtubule (MT) interactions are not required for quiescent-cell nuclear microtubule (Q-nMT) bundle maintenance and mutants affected for kinetochore–kinetochore interactions have reduced viability in quiescence.

**Figure supplement 1—source data 1.** Excel file containing the nuclear MT length distribution in WT and ipl1-1 cells.

**Figure supplement 1—source data 2.** Excel file containing the nuclear MT length distribution in indicated strains background.

**Figure supplement 1—source data 3.** Excel file containing the Bim1 intensity as a function of Tub1 intensity in individual Q-nMT bundles.

**Figure supplement 1—source data 4.** Uncropped western blot for *Figure 3—figure supplement 1E*.

**Figure supplement 1—source data 5.** Uncropped western blot for *Figure 3—figure supplement 1I*.

**Figure supplement 1—source data 6.** Raw western blot for *Figure 3—figure supplement 1E* – Ade13.

**Figure supplement 1—source data 7.** Raw western blot for *Figure 3—figure supplement 1E* – myc.

**Figure supplement 1—source data 8.** Raw western blot for *Figure 3—figure supplement 1I* – csm1 Act1.

**Figure supplement 1—source data 9.** Raw western blot for *Figure 3—figure supplement 1I* – csm1 GFP.

**Figure supplement 1—source data 10.** Raw western blot for *Figure 3—figure supplement 1I* – lsr4 Act1.

**Figure supplement 1—source data 11.** Raw western blot for *Figure 3—figure supplement 1I* – lsr4 GFP.

**Figure supplement 1—source data 12.** Raw western blot for *Figure 3—figure supplement 1I* – mam1 ase13.

**Figure supplement 1—source data 13.** Raw western blot for *Figure 3—figure supplement 1I* – mam1 GFP.

Finally, we investigated whether kinetochore–MT interactions were required for the stability of Q-nMT bundle once it is formed. In 5-day-old WT cells, we used a UV pulsed laser to break the Q-nMT bundle into two pieces. In most cases, the length of both the released fragment and the fragment attached to the SPB remained constant, with the presence of dynamic cMTs after laser pulse testifying for cell viability (*Figure 3I*, *Figure 3—figure supplement 1K*). In contrast, as expected from previous studies (*Khodjakov et al., 2004*; *Zareiesfandabadi and Elting, 2022*), in proliferating cells, dynamic anaphase spindles promptly disassembled after the laser-induced breakage (*Figure 3—figure supplement 1L*). This indicates that once Q-nMT bundles are formed, they do not require MT–kinetochore interaction to be maintained. Accordingly, inactivation of Ndc80 (*Figure 3—figure supplement 1B*) or Ipl1 (*Figure 3—figure supplement 1M*) in cells that were already in quiescence had no effect on

the Q-nMT bundle maintenance. Thus, once formed, Q-nMT stability is established and maintained throughout its length.

## Each phase of Q-nMT bundle formation requires specific kinesins

To further dissect the molecular mechanism of Q-nMT bundle formation, we focused on kinesins. Kar3 is a kinesin that, in complex with its regulator Cik1, can generate parallel MT bundles from an MTOC both in vitro and in proliferative cells (*Manning et al., 1999*; *Mieck et al., 2015*; *Molodtsov et al., 2016*). In quiescence, we found that Kar3-3GFP localized to the SPB, but also as dots along the Q-nMT bundle (*Figure 4A*). In 4-day-old *kar3Δ* cells, the majority of the detected MT structures were extremely short compared to WT (*Figure 4B*; for control without Noc, see *Figure 4—figure supplement 1A*). Similar results were obtained in *cik1Δ* but not in *vik1Δ* cells, which lack the alternative Kar3 regulators (*Figure 4B*). Thus, the Kar3/Cik1 complex is required for phase I.

Cin8 is a kinesin-5 that cross-links MTs (*Bodrug et al., 2020*; *Pandey et al., 2021*; *Singh et al., 2018*), and as such, could play a role in stabilizing Q-nMT bundles. Cin8 was barely detectable in quiescent cells (*Figure 4—figure supplement 1B*) and *cin8Δ* cells assembled Q-nMT bundles that were thinner than in WT cells (*Figure 4C–E*). However, these thinner bundles were stabilized in phase III (*Figure 4C*, *Figure 4—figure supplement 1C and D*). Taken together, these results strongly suggest that Cin8 is required for MT nucleation and/or elongation in phase II but not for MT bundle stabilization in phase III.

Yeast have an additional kinesin-5 called Kip1 (*Fridman et al., 2013*). In quiescence, most of the Kip1-GFP signal was observed at the Q-nMT bundle +end (*Figure 4F*). In *kip1Δ* cells, phase I was slightly delayed. In phase III, while as thick as WT Q-nMT bundles, MT bundles detected in *kip1Δ* cells were not fully stabilized (*Figure 4G and H*, *Figure 4—figure supplement 1E*). These data demonstrate that Kip1 is required for Q-nMT bundle stabilization in phase III. Of note, in phases I and II, Q-nMT bundles were slightly longer in cells deleted for the kinesin-8 Kip3, an MT depolymerase (*Fukuda et al., 2014*, see *Figure 4H*) but have a WT morphology in cells deleted for Kip2, a kinesin that stabilizes cytoplasmic MTs (*Hibbel et al., 2015* and *Figure 4—figure supplement 1F*). Taken together, our results demonstrate that each phase of Q-nMT bundle assembly involves specific kinesins.

## SPB duplication/separation requires Q-nMT bundle disassembly

Finally, we questioned the molecular mechanism of Q-nMT bundle disassembly upon exit from quiescence. We first found that cycloheximide prevented Q-nMT bundle disassembly, indicating that this process requires de novo protein synthesis (*Figure 5A*). Then, we measured Q-nMT bundle thickness upon depolymerization, and found that while Q-nMT bundles shortened, they did not become thinner (*Figure 5B*). In agreement, Nuf2-GFP clusters found at the Q-nMT bundle +end moved back to the SPB while the Nuf2-GFP clusters localized along the Q-nMT bundle remained immobile until they are reached by the +end-associated clusters (*Figure 5—figure supplement 1A*). This shows that not all MT +ends started depolymerizing at the same time, and that longer MTs depolymerized first. This finding is consistent with our model in which MTs are cross-linked along the entire length of the Q-nMT bundle.

Since Kip3 is the only well-characterized yeast MT depolymerase, we examined Q-nMT bundle disassembly in *kip3Δ* cells and found that it was drastically delayed (*Figure 5C and D*, *Figure 5—figure supplement 1B*), yet the overall depolymerization rate remained unaffected (*Figure 5—figure supplement 1B*, left panel). In fact, we found that in WT cells Kip3 needed to be resynthesized upon exit from quiescence (*Figure 5—figure supplement 1C*). We searched for proteins that might be involved in Q-nMT bundle disassembly together with Kip3. Among proteins required for mitotic spindle disassembly, we found that inactivation of *CDC14*, *CDC15*, *CDH1*, *DCC1*, and *TOR1* had no effect on Q-nMT bundle disassembly, as did the inactivation of Ipl1 or proteins involved in kinetochore–MT attachment (*Figure 5—figure supplement 1D–G*). The only additional actor we found was She1, a dynein regulator (*Bergman et al., 2012*), whose deletion strongly exacerbated the *kip3Δ* phenotype (*Figure 5C*, *Figure 5—figure supplement 1H*). Thus, Q-nMT bundle disassembly does not rely on the canonical mitotic spindle disassembly machinery, but rather, specifically requires Kip3 and She1.

Importantly, when we followed quiescence exit at the single-cell level, depolymerization of the Q-nMT bundle always preceded SPB duplication/separation (*Figure 5A*) in both WT and in *kip3Δ*

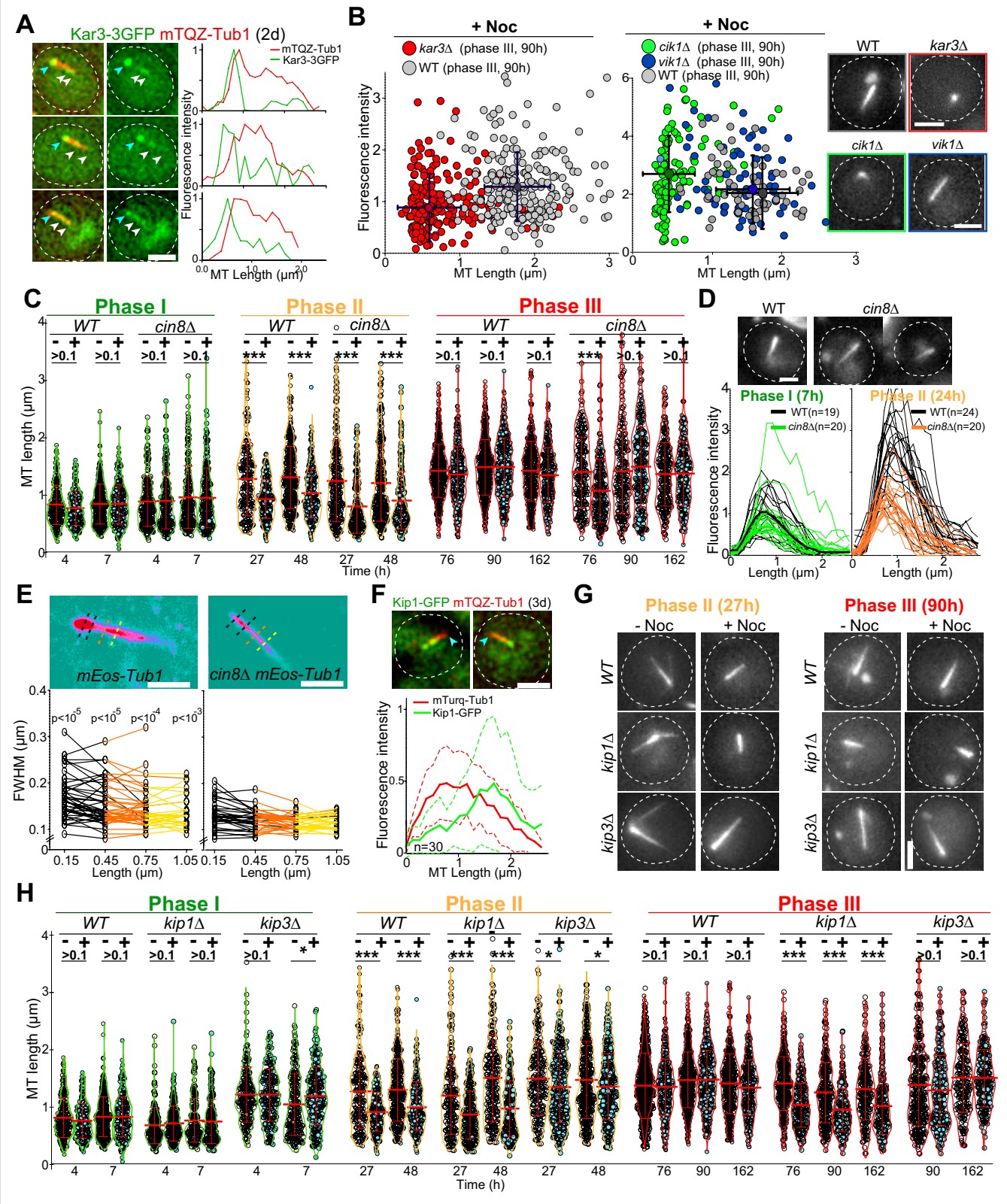

**Figure 4.** Each phase of quiescent-cell nuclear microtubule (Q-nMT) formation requires a specific kinesin. (**A**) Images of representative WT cells (2 d) expressing Kar3-3GFP (green) and mTQZ-Tub1 (red) and corresponding fluorescence intensity along normalized Q-nMT bundles (see 'Materials and methods' section). Bar is 2 μm. (**B**) Morphometric Q-nMT bundle properties distribution in 4 d WT (gray), *kar3Δ* (red), *vik1Δ* (blue), and *cik1Δ* cells (green) expressing mTQZ-Tub1 after Noc treatment (30 μg/ml) – see *Figure 4—figure supplement 1A* for the control without Noc. Each circle corresponds to

*Figure 4 continued on next page*

*Figure 4 continued*

an individual Q-nMT bundle (N ≥ 2, n > 50). Black crosses are mean and SD. Images of representative cells are shown. Bar is 2 µm. (**C**) Nuclear MT length distribution in WT and *cin8Δ* cells expressing mTQZ-Tub1 treated (+) or not (-) with 15 min Noc (30 µg/ml). Each circle corresponds to the length of an individual MT structure, mean and SD are shown. A Student's test (*t*-test with two independent samples) was used to compare (+) or (-) Noc samples from the same experiment (n > 250, N = 1). ***p-value<$1.10^{-5}$. (**D**) Fluorescence intensity along Q-nMT bundles in WT and *cin8Δ* cells expressing mTQZ-Tub1 7 hr and 24 hr after glucose exhaustion. Thin line: intensity from an individual cell; bold line: mean intensity. Images of representative cells are shown. Bar is 2 µm. (**E**) WT and *cin8Δ* cells expressing mEOS3.2-Tub1 were imaged using PALM Images in pseudo-colors of representative cells are shown. Full width at half maximum (FWHM) was measured at the indicated distance from the spindle pole body (SPB). Each line in the bottom graphs corresponds to a single cell (n > 25); p-value between WT and *cin8Δ* is indicated (unpaired *t*-test). Bar is 1 µm. (**F**) Images of representative WT cells (3 d) expressing Kip1-GFP (green) and mTQZ-Tub1 (red). Graphs show fluorescence intensity along normalized Q-nMT bundles (plain and dash lines: mean and SD respectively, n = 30). Bar is 2 µm, blue arrowhead: SPB. (**G**) Representative images of WT, *kip1Δ*, and *kip3Δ* cells expressing mTQZ-Tub1 treated (+) or not (-) with 15 min Noc (30 µg/ml). Bar is 2 µm. (**H**) Nuclear MT length distribution in WT, *kip1Δ*, and *kip3Δ* cells expressing mTQZ-Tub1 treated (+) or not (-) 15 min with Noc (30 µg/ml). Legend is the same as in (**C**); n > 250, N = 1; *p-value<$1.10^{-3}$; ***p-value<$1.10^{-6}$. MT mean length and SD are indicated.

The online version of this article includes the following source data and figure supplement(s) for figure 4:

**Source data 1.** Excel file containing the morphometric Q-nMT bundle property distribution in indicated cells, treated with nocodazole.

**Source data 2.** Excel file containing the nuclear MT length distribution in indicated cells.

**Source data 3.** Excel file containing fluorescence intensity measurement along Q-nMT bundles in WT and cin8Δ cells.

**Source data 4.** Excel file containing individual mesurments of the FWHM mesured along the Q-nMT bundle.

**Source data 5.** Excel file containing individual line scans of Kip1-GFP fluorescence intensities along Q-nMT bundles.

**Figure supplement 1.** The impact of kinesin deletion on quiescent-cell nuclear microtubule (Q-nMT) bundle morphometric parameters.

**Figure supplement 1—source data 1.** Excel file containing fluorescence intensity measurement along Q-nMT bundles in the indicated cells.

**Figure supplement 1—source data 2.** Excel file containing the morphometric Q-nMT bundle property distribution in WT and cin8Δ cells.

cells (*Figure 5—figure supplement 1B*, right panel). When we severely impeded Q-nMT bundle disassembly using *kip3Δ she1Δ*, we found that SPB separation was greatly delayed (*Figure 5E and F*, *Figure 5—figure supplement 1I–K*). Besides, the proliferation rate of *kip3Δ she1Δ* cells was the same as WT cells (*Figure 5—figure supplement 1I*), and, upon quiescence exit, these mutant cells disassembled actin bodies as fast as WT cells (*Figure 5—figure supplement 1K*), demonstrating that *kip3Δ she1Δ* cells were not impaired in sensing quiescence exit signals. These observations suggest that upon exit from quiescence the Q-nMT bundle has to be disassembled prior to SPB duplication/ separation.

## Discussion

Our results shed light on the precise temporal sequences that generate a Q-nMT bundle upon quiescence establishment in yeast. From bacteria to human stem cells, changes in cell physicochemical properties are known to accompany the establishment of quiescence. Among these modifications, cellular volume reduction increases molecular crowding (*Joyner et al., 2016*) and pH acidification both increases viscosity and causes changes in macromolecule surface charges (*Charruyer and Ghadially, 2018*; *Jacquel et al., 2021*; *Munder et al., 2016*; *Persson et al., 2020*). Much evidence suggests that these modifications act as triggers for the auto-assembly of several types of enzyme-containing granules (*Munder et al., 2016*; *Petrovska et al., 2014*; *Rabouille and Alberti, 2017*) or complex structures such as P-bodies and Proteasome Storage Granules (*Peters et al., 2013*; *Jacquel et al., 2021*; *Currie et al., 2023*). Recently, Molines and colleagues demonstrated that cytoplasmic viscosity modulates MT dynamics in vivo (*Molines et al., 2022*). Here, we showed that the formation of the Q-nMT bundle does not depend on changes in the physicochemical properties that cells experience at the onset of quiescence establishment (*Figure 1—figure supplement 1*). In fact, dynamic cMTs and a stable Q-nMT bundle can be observed simultaneously in a quiescent cell (*Figure 1—figure supplement 1I*), while physicochemical properties evolve in parallel in both the nucleus and the cytoplasm (*Joyner et al., 2016*). Importantly, cellular volume reduction, pH acidification, and increased viscosity are observed within minutes upon glucose starvation (*Joyner et al., 2016*), whereas the Q-nMT bundles require a couple of days to be fully assembled (*Figure 1*). Furthermore, Q-nMT bundle formation cannot be delayed in time (*Figure 1H*), and thus likely depends on a transient signal emitted upon

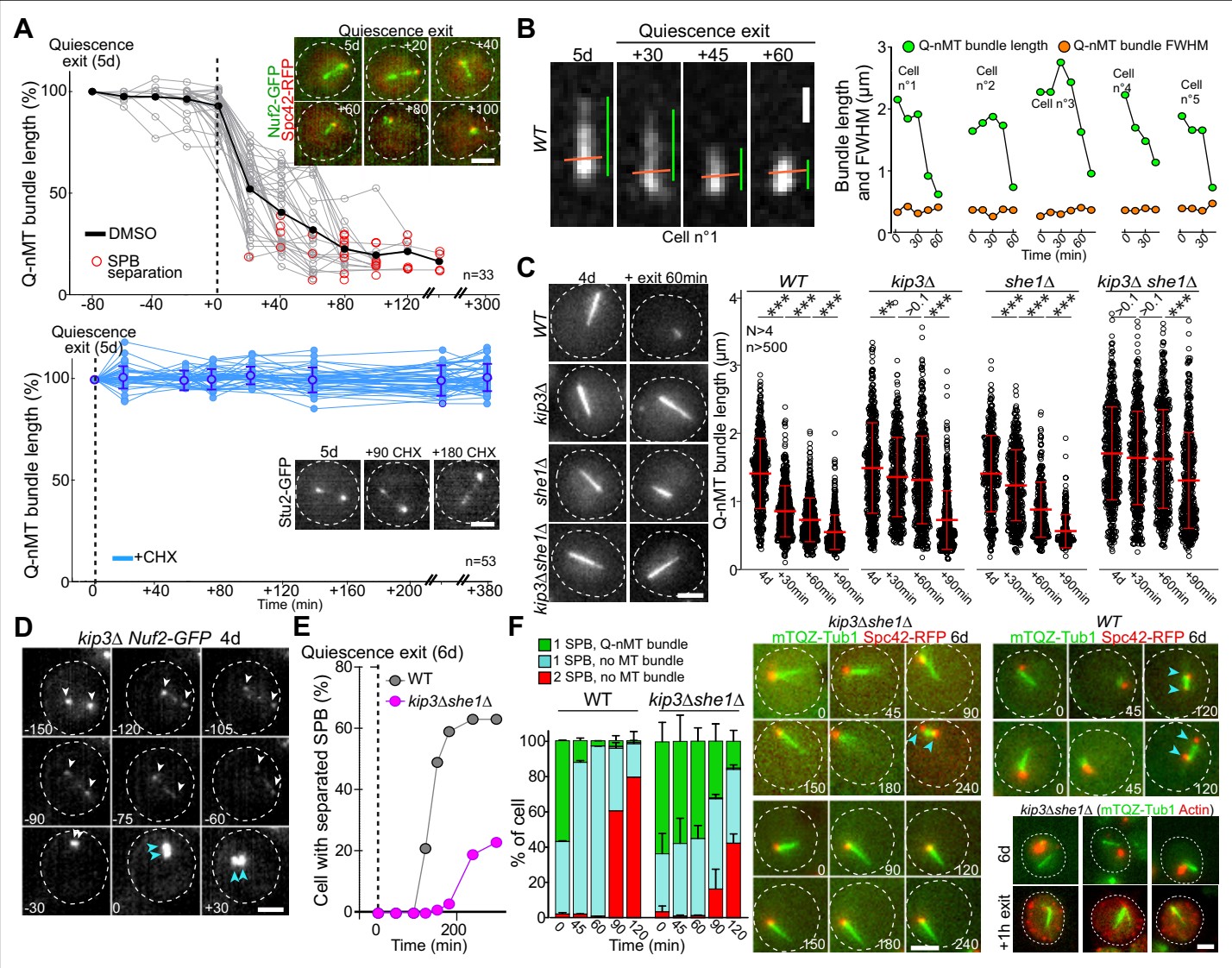

**Figure 5.** Quiescent-cell nuclear microtubule (Q-nMT) bundle disassembly always occurs before spindle pole body (SPB) separation upon quiescence exit. (**A**) WT cells expressing Spc42-RFP (red) (5 d) were re-fed on a YPDA microscope pad. Individual Q-nMT bundles were measured in cells expressing Nuf2-GFP-treated DMSO (top panel, n = 33) or in cells expressing Stu2-GFP treated with CHX (bottom panel, n = 53). Each line corresponds to an individual cell. In the upper panel, time was set to zero at the onset of MT bundle depolymerization. In the lower panel, time was set to zero when cells were deposited on the agarose pad. Images of representative cells are shown. Bar is 2 μm. (**B**) Q-nMT bundle length (green) and fluorescence intensity at full width half maximum (FWHM, orange) were measured upon quiescence exit in WT cells (5 d) expressing mTQZ-Tub1. Representative example of shrinking Q-nMT bundle is shown on the left. Bar is 1 μm. (**C**) Cells of the indicated genotype expressing mTQZ-Tub1 were grown for 4 d, and re-fed. Q-nMT bundle length was measured at the indicated time points, 15 min after a Noc treatment (30 μg/ml), to remove dynamic cMTs that assemble upon quiescence exit. Each circle corresponds to a single cell. MT mean length and SD are indicated. A Student's test (*t*-test with two independent samples) was used to compare samples from the same experiment (n > 250, N = 4). The indicated p-values are the highest p-value calculated among the four repeated experiments. \*\*\*p-value<$1.10^{-5}$. MT mean length and SD are indicated. Images of representative 4-day-old cells of the indicated genotype expressing mTQZ-Tub1 before and 60 min after quiescence exit are shown. Bar is 2 μm. (**D**) Representative images of a *kip3Δ* cell expressing Nuf2-GFP upon quiescence exit. Blue and white arrowheads: SPBs and Q-nMT bundle extremities, respectively. Bar is 2 μm. (**E**) Percentage of 6-day-old WT or *kip3Δshe1Δ* cells expressing Spc42-mRFP1 with separated SPB as a function of time upon quiescence exit (n > 200). (**F**) WT and *kip3Δshe1Δ* cells (6 d) expressing Spc42-mRFP1 (red) and mTQZ-Tub1 (green) were re-fed on a YPDA microscope pad. Percentage of cells with a single SPB with or without Q-nMT bundle or with duplicated SPBs were scored (N = 4, n > 200), SDs are indicated. Images of representative cells are shown. Bar is 2 μm. Bottom-right panel: actin (phalloidin staining, red) in *kip3Δ she1Δ* cells (6 d) expressing mTQZ-Tub1 (green) before and 1 hr after quiescence exit.

The online version of this article includes the following source data and figure supplement(s) for figure 5:

**Source data 1.** Excel file containing Q-nMT bundle length measurement upon quiescence exit.

**Source data 2.** Excel file containing Q-nMT bundle length measurement at the indicated time.

*Figure 5 continued on next page*

*Figure 5 continued*

**Source data 3.** Excel file containing the percentage of cells with a single or a duplicated SBP, and with or without Q-nMT bundle.

**Figure supplement 1.** Quiescent-cell nuclear microtubule (Q-nMT) bundle disassembly in mutants involved in mitotic spindle dismantlement.

**Figure supplement 1—source data 1.** Excel file containing Q-nMT bundle length measurements in WT and kip3Δ cells upon quiescence exit.

**Figure supplement 1—source data 2.** Uncropped western blot for *Figure 5—figure supplement 1C*.

**Figure supplement 1—source data 3.** Raw western blot for *Figure 5—figure supplement 1C* – loading control.

**Figure supplement 1—source data 4.** Raw western blot for *Figure 5—figure supplement 1C*.

glucose depletion. We speculate that the SPB may act as a platform that integrates this signal and transfers it to the MT machinery in order to assemble a stable MT structure.

As it is required for the onset of Q-nMT bundle formation (*Figure 3B*), the CPC component Aurora B/Ilp1, a kinase that plays a key role at the MT–kinetochore interface in response to the tension status, could be one of the targets of the aforementioned nutritional signal. In parallel, upon quiescence establishment, chromosome hyper-condensation (*Guidi et al., 2015*; *Rutledge et al., 2015*) could modify the tension at the kinetochore/MT interface and as such could contribute to the initiation of phase I (*Figure 6b*). Besides, Bim1 has also been implicated in MT–kinetochore attachment (*Dudziak et al., 2021*; *Akhmanova and Steinmetz, 2015*). Although its deletion alone has no effect on Q-nMT bundle formation, its role becomes critical for phase I when kinetochore–MT interactions are already destabilized by the absence of other CPC components, other than Ipl1, such as Bir1 (*Figure 3D*). Taken together, these observations suggest that kinetochore–MT interactions are essential for phase I initiation, as confirmed by the absence of Q-nMT bundle in the *ndc80-1* mutant (*Figure 3A*).

In phase I, SPB-anchored MTs elongate and are concomitantly stabilized (*Figure 1A–C*). This step requires the kinesin-14 Kar3 and its regulator Cik1 (*Figure 4B*). During mating (*Molodtsov et al.,*

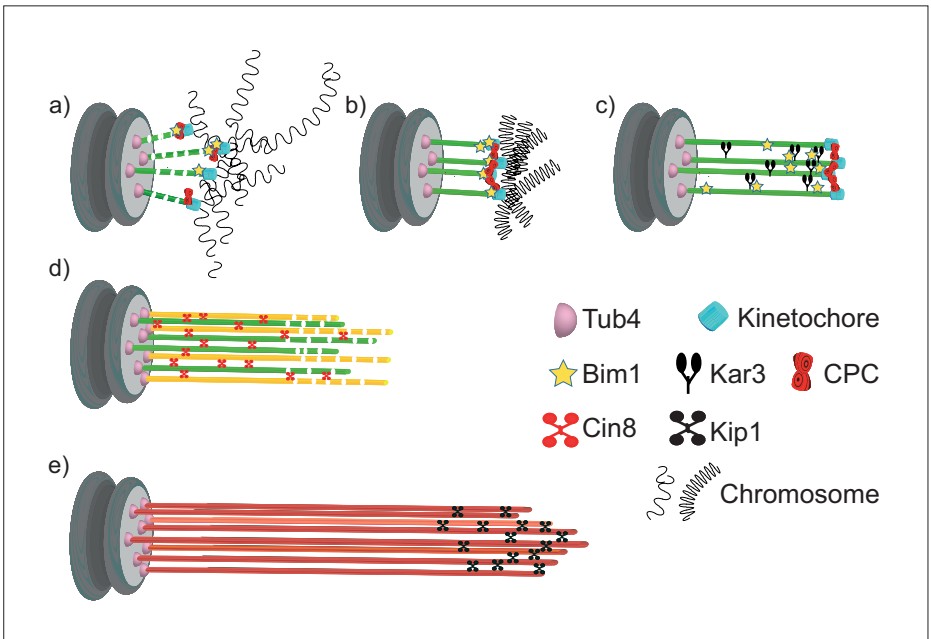

**Figure 6.** Model for quiescent-cell nuclear microtubule (Q-nMT) bundle assembly. (**a**) In G1, the nucleus is in a Rabl-like configuration. (**b**) Upon quiescence establishment, chromosomes get condensed. MT–kinetochore interaction and Ilp1 are required for the onset of phase I. (**c**) Kar3 and its regulator Cik1 are essential to initiate Q-nMT bundle elongation. Although deletion of *BIM1* has no effect, it becomes critical for phase I if kinetochore–MT interactions are destabilized by the absence of Chromosome Passenger Complex components. Kinetochore clustering by the monopolin complex and Slk19 is needed to maintain MT bundling while phase I-MTs elongate. During phase I, Tub4 accumulates at the spindle pole body (SPB). (**d**) In phase II, a second wave of MT nucleation and elongation occurs. Phase II-MTs are concurrently stabilized along pre-existing phase I-MTs, in a Cin8-dependent manner. Phase I and phase II MTs +ends (>1 µm) remain dynamic until the full-length Q-nMT bundle stabilization is reached via the action of Kip1, about 2 d after glucose exhaustion (**e**).

*2016*), and in early mitosis, when half-spindles form (*Kornakov et al., 2020*), Kar3/Cik1, together with Bim1, align and cross-link growing MTs along existing MTs, thereby promoting the organization of MTs into parallel bundles. In mitosis, however, the kinesin-14/EB1 complex promotes MT dynamics (*Kornakov et al., 2020*). It remains to be determined whether the robust MT stabilization in phase I depends on the modification of the kinesin-14/EB1 complex properties and/or on additional specific MT cross-linker(s) (*Figure 6c*).

In quiescence, deletion of either monopolin or *SLK19* results in the formation of short, shattered and flared MT structures, phenotypes that are exacerbated by the deletion of *BIM1* (*Figure 3F–H*). Since monopolin and Slk19 are involved in kinetochore clustering (*Plowman et al., 2019*; *Rabitsch et al., 2003*; *Tóth et al., 2000*; *Lee et al., 2004*; *Mittal et al., 2019*; *Movshovich et al., 2008*; *Richmond et al., 2013*; *Zeng et al., 1999*), we speculate that kinetochore–kinetochore interaction may not only help prevent depolymerization of individual MTs, but also constrain elongating phase I-MTs and facilitate their concomitant cross-linking (*Figure 6d*).

With the initiation of phase I, Tub4 begins to accumulate at the SPB (*Figure 1E*, *Figure 1—figure supplement 1F*) to enable the second wave of MT nucleation observed in phase II. As phase II-MTs elongate, they are simultaneously stabilized along pre-existing phase I-MTs (*Figure 6d*). This step relies on the kinesin-5/Cin8 as Q-nMT bundles assembled in *cin8Δ* cells are thinner than WT Q-nMT bundles (*Figure 4C–E*). Intriguingly, in vitro, the monomeric form of the vertebrate kinesin-5, Eg5, promotes MT nucleation and stabilizes lateral tubulin–tubulin contacts (*Chen et al., 2019*). We assume that in the absence of Cin8 either nucleation of phase II-MTs cannot occur, or phase II-MTs can elongate, but cannot be cross-linked and stabilized to pre-existing phase I-MTs as they grow, and thus rapidly depolymerize. Absence of bundle thickening in phase II is also observed in the '*Tub1-only*' mutant, that is, in the absence of Tub3 (*Figure 2D*). Tubulin isotypes or tubulin PTMs are known to promote the recruitment of specific kinesins and thereby modify MT properties (e.g., see *Sirajuddin et al., 2014*; *Peris et al., 2009*; *Dunn et al., 2008*). Could Tub3 be important for Cin8 recruitment or function during phase II?

Following phase II, MTs elongate, but their distal parts (>1 μm) are not yet stable (*Figure 1A*). Stabilization of full-length Q-nMT bundles is achieved approximately 4 d after glucose exhaustion. This step depends on the kinesin-5/Kip1 (*Figure 4G and H*, *Figure 4—figure supplement 1E*), which is essential for cross-linking and stabilizing elongating MTs during phase III (*Figure 6e*). Accordingly, we observed a Kip1 enrichment at the Q-nMT bundle +ends (*Figure 4F*). Thus, consistent with their ability to form homo-tetramers capable of cross-linking (*Acar et al., 2013*; *Kapitein et al., 2005*; *Scholey et al., 2014*; *Weinger et al., 2011*) and stabilizing parallel MTs (*Kapitein et al., 2005*; *Shimamoto et al., 2015*; *Yukawa et al., 2020*), and in addition to their role of sliding anti-parallel MTs, the two yeast kinesin 5 are essential for Q-nMT bundle formation. However, in quiescence, as well as in mitosis, Kip1 and Cin8 have non-equivalent functions (*Roostalu et al., 2011*; *Shapira and Gheber, 2016*).

Thus far, our results indicate that the mechanism of Q-nMT bundle formation is distinct from that of mitotic spindle assembly. Similarly, we show that Q-nMT bundle disassembly does not involve the same pathway as mitotic spindle disassembly as it requires Kip3 (*Figure 5C*) but is independent of the anaphase-promoting complex and Aurora B/Ipl1 (*Figure 5—figure supplement 1D–G*). Interestingly, the longest MTs begin to depolymerize from their +ends first. Then, when they reach the +ends of shorter MTs, the shorter ones proceed to depolymerize alongside them (*Figure 5B*, *Figure 5—figure supplement 1A*). This cooperative behavior has been observed in vitro within parallel MT bundles (*Laan et al., 2008*). Our data demonstrate that such a comportment does exist in vivo.

Finally, we show that the Q-nMT bundle is required for the survival of WT cells in quiescence (*Figure 3—figure supplement 1H*). While we do not yet know why this structure is important for chronological aging, it is clear that the presence of the Q-nMT bundle modifies the organization of the nucleus (*Laporte et al., 2013*; *Laporte and Sagot, 2014*; *Laporte et al., 2016*). Since chromatin organization influences gene expression, one hypothesis could be that the presence of the Q-nMT bundle is required for the expression of genes necessary for cell survival in quiescence. Another attractive hypothesis is that the Q-nMT bundle could be the yeast analog of the mammalian primary cilium. Indeed, these two structures are formed upon proliferation cessation, they are both composed of highly stable parallel MTs, their formation involves MT motors, and they are both templated from the centriole/SPB. More importantly, we show here that Q-nMT bundle disassembly always occurs

prior to SPB separation upon exit from quiescence (*Figure 5E and F*, *Figure 5—figure supplement 1J*). Thus, the Q-nMT bundle disassembly may control reentry into the proliferation cycle, just as the primary cilium does in ciliated mammalian cells (*Goto et al., 2017*; *Kim and Tsiokas, 2011*).

## Materials and methods

### Yeast strains, plasmids, and growth conditions

All the strains used in this study are isogenic to BY4741 (*mat a, leu2Δ0, his3Δ0, ura3Δ0, met15Δ0*) or BY4742 (*mat alpha, leu2Δ0, his3Δ0, ura3Δ0, lys2Δ0),* available from GE Healthcare Dharmacon Inc (UK), except for *Figure 2*, in which strains of the S288c background were used (*Nsamba et al., 2021*), and *Figure 1—figure supplement 1F*, in which the W303 background was used. BY strains carrying GFP fusions were obtained from Thermo Fisher Scientific (Waltham, MA). Integrative plasmids p*Tub1-mTurquoise2-Tub1*, p*Tub1-wasabi-Tub1*, p*Tub1-mRUBY2-Tub1*, and p*Tub1-mEOS2-Tub1* were a generous gift from Wei-Lih Lee (*Markus et al., 2015*). The strain expressing SPC42-mRFP1 is a generous gift from E. O'Shea (*Huh et al., 2003*). The strain expressing TUB4-mTQZ is a generous gift from S. Jaspersen (*Burns et al., 2015*). The *ipl1-5as* is a generous gift from A. Marston (*Nerusheva et al., 2014*). *Ipl1-1*, *ndc80-1,* and *sli15-3* are generous gifts from C. Boone. A copy of tDimer (RFP) was integrated at the 3′ end of *TUB3* endogenous loci. Three copies of eGFP in tandem were integrated at the 3′ end of *DAD2*, *TUB3*, *MAM1*, or *KAR3* endogenous loci, respectively. Plasmids for expressing Nup2-RFP (p3695) or Bim1-3xGFP (p4587) from the endogenous locus were described in *Laporte et al., 2013*.

For *Figure 1F* and *Figure 1—figure supplement 1H*, *pHIS3:mTurquoise2-Tub1+3′UTR::LEU2* was integrated at *TUB1* locus, then a pRS303-*ADH2p-mRuby2-Tub1* was integrated at the *HIS3* locus. To generate this plasmid, the *ADH2* promoter was amplified from yeast genomic DNA flanked with NotI and SpeI restriction sites and inserted in pRS303. The mRuby2-Tub1+3′UTR, including the 116-nucleotide intron and 618 nucleotides downstream the stop codon, was cloned between the SpeI and SalI sites.

Yeast cells were grown in liquid YPDA medium at 30°C in flasks as described previously in *Sagot et al., 2006* except for experiments using thermosensitive strains, where cells were first grown at 25°C then shifted at 37°C for the indicated time before imaging.

Experiment in *Figure 1—figure supplement 1J* was performed as described in *Orij et al., 2009* and *Peters et al., 2013*. In brief, proliferating cells at an $OD_{600nm}$ of 0.5 were transferred in HEPES buffer (25 mM HEPES, pH 7.4, 200 mM KCl, 1 mM $CaCl_2$, and 2% dextrose), buffered either at pH 4 or 7.5, in the presence of 100 µM CCCP (Sigma-Aldrich). After 150 min at 30°C with shaking, cells were imaged (for mTQZ-Tub1) or fixed with formaldehyde and stained using Alexa Fluor 568-phalloidin (Invitrogen) as described (*Sagot et al., 2006*).

For live-cell imaging, 2 µl of the cell culture were spotted onto a glass slide and immediately imaged at room temperature (RT).

For quiescence exit (*Figure 5*), cells were first incubated 2–3 min in liquid YPD and then 2 µl were spread onto a 2% agarose microscope pad containing YPD. Individual cells were imaged every hour, up to 6 or 12 hr at 21°C. For quiescence exit in the presence of cycloheximide (*Figure 5A*), cells were preincubated for 30 min in the presence of the drug prior to quiescence exit.

Immunofluorescence was done as in *Laporte et al., 2016*, and Alexa Fluor Phalloidin (Invitrogen) staining as in *Sagot et al., 2006*.

Cycloheximide was used at 180 µM (Sigma-Aldrich), nocodazole was used at 30 µg/ml (7.5 µM) (Sigma-Aldrich), and NA-PP1 was used at 50 µM (Sigma-Aldrich).

### Fluorescence microscopy

Cells were observed on a fully automated Zeiss 200M inverted microscope (Carl Zeiss, Thornwood, NY) equipped with an MS-2000 stage (Applied Scientific Instrumentation, Eugene, OR), a Lambda LS 300 W xenon light source (Sutter, Novato, CA), a 100×1.4 NA Plan-Apochromat objective, and a 5-position filter turret. For RFP and mRuby2 imaging, we used a Cy3 filter (excitation: HQ535/50 nm; emission: HQ610/75 nm; beam splitter: Q565 nm lp). For GFP imaging, we used an FITC filter (excitation: HQ487/25 nm; emission: HQ535/40 nm; beam splitter: Q505 nm lp). For mTurquoise2 imaging, we used a DAPI filter (excitation: 360/40 nm; emission: 460/50 nm; beam splitter: 400 nm). All the

filters are from Chroma Technology Corp. Images were acquired using a CoolSnap HQ camera (Roper Scientific, Tucson, AZ). The microscope, camera, and shutters (Uniblitz, Rochester, NY) were controlled using SlideBook software 5.0 (Intelligent Imaging Innovations, Denver, CO).

For PALM (*Figure 4E*), a Nikon Ti-Eclipse equipped with iLas2 TIRF arm, laser diodes (405, 488, 532, 561, 642 nm), a 100×1.49 oil (TIRF) objective connected to an EMCCD Camera Photometrics Evolve was used.

For expansion microscopy (*Figure 2—figure supplement 1A* and *Figure 3—figure supplement 1A*), spheroplasts were obtained as described in *Laporte et al., 2016*. After washing in 10 mg/ml NaBH$_4$ in PEMS for 10 min, spheroplasts were seeded on 12 mm round poly-L-lysine coated coverslips, washed twice for 30 min in 100 µl PEM-BAL (PEM +1% BSA), and incubated with the primary antibodies (anti-GFP from mouse; Roche, ref 11814460001, 1/50; anti-α-tubulin from rat; YOL1/34, Abcam, ref Ab6161, 1/100) for 1 hr at 37°C. Cells were then washed three times in PEM-BAL and incubated with the secondary antibodies (goat anti-mouse Alexa Fluor 488 1/200 and donkey anti-rat Alexa Fluor555; A11029 and A21434, respectively, Thermo Fisher Scientific) for 45 min at 37°C. Cells were then washed three times with PBS. Cells were processed for Expansion Microscopy as previously described in *Bahri et al., 2021*. Spheroplasts were incubated for 10 min in 0.25% GA in PBS, washed in PBS three times for 5 min, and processed for gelation. A drop of 100 µl of ExM MS (8.625% [wt/wt] SA, 2.5% [wt/wt] AA, 0.15% [wt/wt] BIS, 2 M NaCl in 1× PBS) was placed on the chilled Parafilm, and coverslips were put on the drop with cells facing the solution and incubated for 3 min. Then the coverslips were transferred to 35 µl of ExM MS supplemented with 0.2% APS and 0.2% TEMED, with the initiator (APS) added last. Gelation proceeded for 3 min on ice, and samples were incubated at 37°C in a humidified chamber in the dark for 1 hr. Then coverslips with attached gels were transferred into a six-well plate for incubation in 2 ml of digestion buffer (1× TAE buffer, 0.5% Triton X-100, 0.8 M guanidine hydrochloride, pH ~8.3) supplemented with DAPI (1 µg/ml) for 10–15 min at 37°C, until the gels detached. Fresh proteinase K at 8 units/ml was then added and samples incubated at 37°C for 30 min. Finally, gels were removed and placed in 10 ml Petri dishes filled with ddH$_2$O for expansion. Water was exchanged at least twice every 30 min and incubated in ddH$_2$O overnight at RT. Gels expanded between 4× and 4.2× according to SA purity. Expanded cells were imaged with a UPlanS Apo 100×/1.4 oil immersion objective in an Olympus IX81 microscope (Olympus, Tokyo, Japan). For structured illumination microscopy (*Figure 3—figure supplement 1A*), a ZEISS Elyra 7 Lattice SIM was used.

Laser ablation (*Figure 3I*, *Figure 3—figure supplement 1A, K, and L*) was performed at RT with a 100× oil Plan-Apochromat objective lens (NA 1.4) and an Axio-Observed.Z1 microscope (Carl Zeiss) equipped with a spinning disk confocal (Yokogawa), an EMCCD Evolve camera (Photometrics and Roper Scientific), and 491 nm (100 mW; Cobalt Calypso) and 561 nm (100 mW; Cobolt Jive) lasers. Images were acquired with Metamorph software (Roper Scientific). Every 30–120 s, a Z-series of 0.4 µm steps were acquired. A 355 nm microchip laser (Teem Photonics) with a 21 kHz repetition rate, 0.8 µJ energy/pulse, 2 kW of peak power, and 400 ps pulse width, powered with an iLas2 PULSE system (Roper Scientific), was used between 10 and 40% power with one pulse of a spot length of 100 points. Breakage was considered successful if a non-alignment between the two remaining Q-n MT bundle fragments was observed.

Z-stacks were deconvolved using the Deconvolution Lab plugin (*Figures 1D and F*, *3C, E, and F*, *4A–F*, *Figure 1—figure supplement 1A and E*, *Figure 2—figure supplement 1A*, and *Figure 4—figure supplement 1B*).

In *Figures 1H and F*, *2B and C*, *3C*, *4D and G*, *Figure 1—figure supplement 1I*, *Figure 2—figure supplement 1E* and *Figure 3—figure supplement 1A*, a fuzzy fluorescence signal was detected in some cell cytoplasm using the GFP long-pass filter set. This signal is not GFP but rather due to a non-specific yellow background fluorescence.

## Image analysis

Distribution and associated statistics were performed using GraphPad Prism 5 (GraphPad Software, Inc, La Jolla) and Excel (Microsoft). Unless specified, a Student's test (*t*-test with two independent samples) was used to compare two conditions. Among p-values obtained for biological repetitions of the same experiment (*Figures 1A*, *2B and C*, *3A, D, F, and H*, *4C and H*, and *5C*), the indicated p-values are the highest calculated among the repeated experiments. p-Values >0.05 are indicated,

while p-values <0.05 are indicated using ** ($0.05 >$ p-value $> 1.10^{-4}$) and *** if the p-values were $< 1.10^{-5}$. In histograms and scatter dot plots, the mean is shown and the error bars indicate SD.

MT length was measured on MAX-projected image using ImageJ. For MT fluorescence intensity measurements (*Figures 1B*, *2D*, *4D*, *Figure 1—figure supplement 1A*, and *Figure 4—figure supplement 1C and E*), a line scan (i1) of 4 pixels width containing both FP signal and background was drawn along MTs on sum projection image (3–4 Z-plans) using ImageJ. A line of 8 pixels width (at the same location) was drawn in order to calculate the intensity of the surrounding background (i2). The real intensity (ir) was calculated as follows: ir = [i1 – ([(i2 × i2$_{surface}$) – (i1 × i1$_{surface}$)]/(i2$_{surface}$ – i1$_{surface}$)) * i1$_{surface}$]. We arbitrarily set the 'zero' at 0,5 µm before the fluorescence increased onset on the SPB side in order to align all intensity measurements. Similarly, to measure Tub4 fluorescence intensity (*Figure 1E*, *Figure 1—figure supplement 1A and F*). two squares (i1) and (i2) were drawn around the fluorescence signal. Intensity was calculated as follows: int = [i1 – ([(i2 × i2$_{surface}$) – (i1 × i1$_{surface}$)]/(i2$_{surface}$ – i1$_{surface}$)) * i1$_{surface}$].

For morphometric Q-nMT bundle property distribution (*Figures 1C*, *4B*, and *Figure 4—figure supplement 1A and D*), the mean fluorescence intensity was measured in each individual cell in a finite area localized adjacent to the SPB (0.4 µm – 3–4 Z-stack sum projected) as an estimate of Q-nMT bundle width and plotted as a function of the corresponding bundle length.

For Nuf2-GFP fluorescence intensity measurement at the SPB (*Figure 1—figure supplement 1E*, bottom panel), GFP and RFP line-scan measurements were done on a four Z-planes sum projection. The 'SPB localization zone' was defined as the length ending 250 µm after the brightest RFP signal of Spc42-RFP. The remaining Q-nMT bundle zone was defined as the '+end zone'. The total Nuf2-GFP signal detected along the Q-nMT bundle was set to 100% to determine the percentage of the signal measured at the 'SPB localization zone' and the '+end zone'.

Normalized Q-nMT bundle length (*Figures 3C* and *4A and F*) was calculated after line-scan intensity measurement. mTQZ-Tub1 intensity slopes were first aligned on their inflexion points. To compare Q-nMT bundles with different lengths, we first sorted Q-nMT bundles <1.8 µm. After an artificial isotropic expansion, we fit all MT structures to the same length. Then, the corresponding mean intensity of Q-nMT bundles (mTZQ-Tub1) and GFP signal were calculated.

To measure Q-nMT bundle depolymerization rate (*Figure 5A*, *Figure 5—figure supplement 1B*), individual Q-nMT bundle lengths were measured over time. The first length measurement was set to 100%. A fluorescence drop above or equal to 25% defines the inflexion point of the slope and was used to align the length measurements for Q-nMT bundles.

The full width at half maximum (FWHM) (*Figure 5B*, *Figure 1—figure supplement 1B*) was calculated by measuring fluorescence intensity of a line crossing perpendicularly a Q-nMT bundle, at 0.5 µm from the SPB, using a sum projection image. After fitting the intensity level with Gaussian distribution and obtaining the standard deviation (σ) value, FWHM was calculated using the equation $2\sqrt{(2\ln2)} \times \sigma$.

## Western blots

Western blots were done as described in *Sagot et al., 2006* using anti-GFP antibodies (Abcam); anti-Tat1 antibodies (a generous gift from J-P. Javerzat) and antibody against the budding yeast Act1 or Sac6, generous gifts from B. Goode, and anti-TagRFP (mRuby2) antibodies, a generous gift from M. Rojo.

## Phenotypical analysis of cells without Q-nMT bundle

WT cells were grown to glucose exhaustion (OD ~6.5) (*Figure 1F*) or for 5 d (*Figure 1—figure supplement 1L*), washed with 'old YPDA' (YPDA medium in which cells were grown for 4 d and then filtered to remove cells) and treated with either DMSO or Noc (30 µg/ml) for 24 hr. Cells were then washed twice in 'old YPDA' before being incubated in the same medium. Dead cells were identified using methylene blue staining. The capacity of cells to exit quiescence (*Figure 1—figure supplement 1L*, right panel) was scored after micro-manipulation of live cells (cells not stained with methylene blue) as described in *Laporte et al., 2011* (N = 4, n > 100).

## Acknowledgements

We thank the Bordeaux Imaging Center for the help in super-resolution imaging. We express our gratitude to E O'Shea, A Marston, S Jaspersen, J-P Javerzat, Wei-Lih Lee, C Boone, and B Goode

for sharing reagents and M Rojo for providing us with the mCherry-specific antibodies. We thank Michaël Gué (Zeiss) for helping us a with the lattice SIM Elyra 7 microscope. We would like to thank J-P Javerzat for helpful and constructive discussions about our work. DL, AML, JD, and IS were supported by a grant from the ANR-21-CE13-0023-01, la Ligue Contre le Cancer Régionale – Dordogne grant #193366 and the CNRS. MG and EN were supported by a National Science Foundation grant number MCB-1846262. AR was supported by the Conseil Régional de Nouvelle Aquitaine (#20111301010) and the CNRS.

## Additional information

### Funding

| Funder | Grant reference number | Author |
| --- | --- | --- |
| Agence Nationale de la Recherche | ANR-21-CE13-0023-01 | Damien Laporte<br>Aurelie Massoni-Laporte<br>Jim Dompierre<br>Isabelle Sagot |
| Ligue Contre le Cancer | #193366 | Damien Laporte<br>Aurelie Massoni-Laporte<br>Isabelle Sagot |
| National Science Foundation | MCB-1846262 | Mohan L Gupta |
| Conseil Regional de Nouvelle Aquitaine | #20111301010 | Anne Royou |

The funders had no role in study design, data collection and interpretation, or the decision to submit the work for publication.

### Author contributions

Damien Laporte, Conceptualization, Formal analysis, Supervision, Funding acquisition, Validation, Investigation, Visualization, Methodology, Writing - original draft, Writing - review and editing; Aurelie Massoni-Laporte, Formal analysis, Investigation, Methodology; Charles Lefranc, Investigation, Methodology; Jim Dompierre, David Mauboules, Investigation; Emmanuel T Nsamba, Mohan L Gupta, Methodology; Anne Royou, Resources; Lihi Gal, Maya Schuldiner, Resources, Investigation, Methodology; Isabelle Sagot, Conceptualization, Supervision, Funding acquisition, Validation, Visualization, Writing - original draft, Project administration, Writing - review and editing

### Author ORCIDs

Damien Laporte (iD) http://orcid.org/0000-0002-1556-5253
Maya Schuldiner (iD) http://orcid.org/0000-0001-9947-115X
Isabelle Sagot (iD) http://orcid.org/0000-0003-2158-1783

Reviewer #1 (Public Review): https://doi.org/10.7554/eLife.89958.3.sa1
Reviewer #2 (Public Review): https://doi.org/10.7554/eLife.89958.3.sa2
Author Response https://doi.org/10.7554/eLife.89958.3.sa3

## Additional files

### Supplementary files

• MDAR checklist

### Data availability

All data generated or analysed during this study are included in the manuscript and supporting files. Numerical data source files have been provided for Figures and Figure supplements.

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
