## [Editor Report · eLife assessment]

This work presents **important** insights regarding the mechanism underlying the assembly, maintenance, and disassembly of a very stable microtubule-based structure, termed quiescent-cell nuclear microtubule (Q-nMT) bundle, which is formed in quiescent yeast cells to ensure cell survival and viability. This insight will help elucidate how very stable microtubules can exist alongside very dynamic microtubules, which is still poorly understood. While the experimental support is overall **solid**, additional analyses using state-of-the-art methodology would further strengthen some of the claims.

---

## [Referee Report · Reviewer #1 (Public Review)]

In their manuscript, Laporte et al. analyze the process of formation of the quiescent-cell nuclear microtubule (Q-nMT) bundle, a set of parallel MTs that emanate from the nuclear side of the spindle pole bodies (SPBs) upon the entry of *Saccharomyces cerevisiae* cells in quiescence. Based on their results, the authors propose that Q-nMT bundle formation is a multistep process that comprises three distinct sequential phases. The authors further evaluate the role of different factors during the growth of the Q-nMT bundle upon quiescence entry, as well as during the disassembly of this structure once the cells resume their proliferation.

The Q-nMT is an interesting cellular structure whose physiological function is still widely unknown. Hence, providing new insights into the dynamics of Q-nMT bundle formation and identifying new factors involved in this process is an interesting topic of relevance in the field. The authors made a substantial effort in order to evaluate Q-nMT bundle establishment and provide a considerable amount of data, mainly obtained from microscopy analyses. Overall, the experiments are mostly well described and properly executed, and the data in the manuscript are clearly presented. Despite the interest of the study, nonetheless, there are several issues that could affect the validity of some conclusions drawn. In this way, regarding the analysis of the dynamics of Q-nMT bundle formation, the described experimental setup raises certain concerns, which mostly derive from the use of the microtubule-depolymerizing agent nocodazole as the only approach to evaluate this process. Also, regarding the factors involved in Q-nMT formation, the differences in microtubule length with the wild type strain, despite being statistically significant, are really subtle for many of the mutants analyzed (e.g., bir1, slk19, etc.). Furthermore, it is also puzzling that an effect on Q-nMT formation is proposed for meiosis-specific factors such as Mam1, which might as well be present during quiescence, but seems to be also detected in proliferating cells. Lastly, the evidence shown are insufficient to provide a direct link between defects in cell viability and aberrant Q-nMT formation.

---

## [Referee Report · Reviewer #2 (Public Review)]

Summary: The authors investigate the assembly of the Q-nMT, a stable microtubule structure that is assembled during quiescence. Notably, the authors show that the formation of the Q-nMT cannot be solely explained by changes in the physico-chemical properties of quiescent cells. The authors report that Q-nMT assembly occurs in three regulated steps and identify kinesin motor proteins involved in the assembly and disassembly of the structure.

Strengths: The findings provide new insight into the assembly and possible function of the Q-nMT with respect to the response of haploid budding yeast to glucose starvation.

Weaknesses: The manuscript would benefit from more precise language and requires additional clarification regarding how claims are supported by the evidence. Clear definitions are also required, for example "active process" is not defined. Some conclusions are not supported by the results, for example the claim that the Q-nMT functions as a checkpoint effector that inhibits re-entry into the cell cycle.

After reviewing the responses of the authors and the revised manuscript I am now satisfied with the study in its current form.

---

## [Author Response]

The following is the authors’ response to the original reviews.

**Reviewer #1 (Recommendations for The Authors):**
To hopefully contribute to more strongly support the conclusions of the manuscript, I am including a series of concerns regarding the experiments, as well as some recommendations that could be followed to address these issues:(1) The Q-nMT bundle is largely unaffected by the nocodazole treatment in most phases during its formation. However, cells were only treated with nocodazole for a very short period of time (15 min). Have the authors analyzed Q-nMT stability after longer nocodazole exposures? Is a similar treatment enough to depolymerize the mitotic spindle? This result could be further substantiated by treatment with other MT-depolymerizing agents. Furthermore, the dynamicity of the Q-nMT bundle could be ideally also assessed by other techniques, such as FRAP.

The experiments suggested by the reviewer have been published in our previous paper (Laporte et al, JCB 2013). In this previous study, we presented data demonstrating the resistance of the Q-nMT bundle to several MT poisons: TBZ, benomyl, MBC (Sup Fig 2D) and to an increasing amount of nocodazole after a 90 min treatment (Sup Fig2E). These published figures are provided below.

**Author response image 1. sa3fig1:** The nMT array contains highly stable MTS. (A) Variation Of nuclear MT length in function Of time (second) in proliferating cells. Cells express GFP•Tubl (green) and Nup2•RFP (red). Bars, 2 pm. N = l, n is indicated. (B) Variation of the nMT array length in function of time measured for BirnlGFP—expressing cells In = 161, for 6-d•old Dad2GFP—expressing cells In = 171, for Stu2GFP—expressing cells (n = 17), and 6•d-old Nuf2• GFP—expressing cells (n = 17). Examples Of corresponding time lapse are shown. Time is in minutes experiments). Bar, 2 pm. (CJ Nuf2•GFP dots detected along nMT array (arrow) are immobile. Several time lapse images of cells are shown. Time is in minutes. gar, 2 pm _ MT organizations in proliferating cells and 4-d•old quiescent cells before and after a 90-min treatment With indicated drugs. Bar, 2 pm. (E) MT organizations in Sci-old quiescent cells before and after a 90min treatment With increasing concentrations Of nocodazole.

In the same article, we showed that Q-nMT bundles resist a 3h nocodazole treatment, while all MT structures assembled in proliferating cells, including mitotic spindle, vanished (see Fig 2E below). In addition, in our previous article, FRAP experiments were provided in Fig 2D.

**Author response image 2. sa3fig2:** The nuclear array is composed of stable MTS. Variation of the length in function of time of (A) aMTs in proliferating cells, (B) nMT array in quiescent cells (7 d), and the two MT structures in early quiescent cells (4 d). White arrows point ot dynamic aMTs. In A—C, N = 2, n is indicated (ID) FRAP on 7-d-old quiescent cells. White arrows point to bleach areas. Error bars are SEM. In A—D. time is in seconds. (E) nMT array is not affected by nocodazole treatment. Before and various times after carbon exhaustion (red dashed line), cells were incubated for 3 h with 22.5 pg/pL nocodozole and then imaged. The corresponding control experiment is shown in Fig I A. In all panels, cells expressing GFP-TtJbl (green) and Nup2-RFP (red) are shown; bars, 2 pm.

This previous study was mentioned in the introduction and is now re-cited at the beginning of the results section (line 107-108).

As expected from our previous study, when proliferating cells were treated with Noc (30 µg/ml) in the same conditions as in Fig1, most of the short and the long mitotic spindles vanished after a 15 min treatment as shown in the graph below.

**Author response image 3. sa3fig3:** Proliferating cells expressing NOf2=GFP and mTQZ-TUb1 (00—2) were treated or not With NOC (30vgfmI) for 15 min. % Of cells With detectable MT and representative cells are shown. Khi-teet values are indicated. Bar: 2 pm,

(2) The graph in Figure 1B is somewhat confusing. Is the X-axis really displaying the length of the MTs as stated in the legend? If so, one would expect to see a displacement of the average MT length of the population as cells progress from phase II to phase III, as previously demonstrated in Figure 1A. Likewise, no data points would be anticipated for those phases in which the MT length is 0 or close to 0. Moreover, when the length of half pre-anaphase mitotic spindle was measured as a control, how can one get MT lengths that are equal or close to 0 in these cells? The length of the pre-anaphase spindle is between 2-4 um, so MT length values should range from 1 to 2 um if half the spindle is measured.

The graph in Fig1B represents the fluorescence intensity (a proxy for the Q-nMT bundle thickness) along the Q-nMT bundle length.

Fluorescence intensity is measured along a “virtual line” that starts 0,5 µm before the extremity of the QnMT bundle that is in contact with the SPB. In other words, we aligned all intensity measurements at the fluorescence increasing onset on the SPB side. We arbitrarily set the ‘zero’ at 0,5um before the fluorescence increased onset. That is why the fluorescence intensity is zero between 0 and 0,5 µm – The X-axis represents this virtual line, the 0 being set 0,5 µm before the Q-nMT bundle extremity on the SPB side. This virtual line allows us to standardize our “thickness” measurements for all Q-nMT bundles.

Using this standardization, it is clear that the length of the Q-nMT bundles increased from phase II to III (see the red arrow). Yet, as in phase II, Q-nMT bundles are not yet stable, their lengths are shorter in phase II than in phase II after a Noc treatment (compare the end of the orange line and the end of the blue line in phase II).

**Author response image 4. sa3fig4:** 

This is now explained in details in the Material and Methods section (line 539-545).

This is the same for the inset of Fig 1B and in Sup Fig 1A, in which we measured fluorescence intensity along the halfmitotic spindle just as we did for MT bundle. The X-axis represent a virtual line along the mitotic spindle, starting 0,5 µm before the SBP spindle extremity.

**Author response image 5. sa3fig5:** 

(3) Microtubules seem to locate next to or to extend beyond the nucleus in the control cells (DMSO) in Figure 1H. Since both nuclear MTs and cytoplasmic MTs emanate from the SPBs, it would have been desirable to display the morphology of the nucleus when possible. Moreover, since the nucleus is a tridimensional structure, it would also be advisable to image different Z-sections.

Analysis demonstrating that Q-nMT bundles are located inside the nucleus have been provided in our previous paper (Laporte et al, JCB 2013). In this article most of the images are maximal projections of Z-stacks in which the nuclear envelope is visualized via Nup2-RFP (see Fig1 of Laporte et al, JCB 2013 as an example below).

**Author response image 6. sa3fig6:** MTsare organized as a nuclear array in quiescent cells. (A) MT reorganization upon quiescence entry. Cells expressing GFP-Tub1 (green) and Nup2RFP (red) are shown. Glucose exhaustion is indicated as a red dashed line. Quiescent cells dl expressing Tub I-RFP and either Spc72GFP,

In Laporte et al, JCB 2013, we also provided EM analysis both in cryo and immune-gold (Fig 1E below).

**Author response image 7. sa3fig7:** (top) or coexpr;sse8 with Tub I-RFP (bottom). Arrows point dot along the nMT array. Bars: (A—C) 2 pm. (E) AMT arroy visualized in WT cells by EMI Yellow arrows, MTS; red arrowheads, nuclear membrane; pink arrow, SPB. Insets: nMT cut transversally. Bar, 100 nm.

(4) Movies depicting the process of Q-nMT bundle formation in live cells would have been really informative to more precisely evaluate the MT dynamics. Likewise, together with still images (Fig 1D and Supp. Fig. 1D), movies depicting the changes in the localization of Nuf2-GFP would have further facilitated the analysis of this process.

In a new Sup Fig 1E, we now provide images of Q-nMT bundle formation initiation in phase I, in which it can be observed that Nuf2-GFP accompanies the growth of MT (mTQZ-TUB1) at the onset of Q-nMT bundle formation. Unfortunately, it is technically very challenging to follow the entire process of Q-nMT bundle formation in individual cells, as it takes > 48h. Indeed, for movies longer than 24h, on both microscope pads or specific microfluidic devices (Jacquel, et al, eLife 2021), phototoxicity and oxygen availability become problematic and affect cells’ viability.

(5) Western blot images displaying the relative protein levels for mTQZ-Tub1 and of the ADH2 promoter-driven mRuby-Tub1 at the different time points should be included to more strongly support the conclusion that new tubulin molecules are introduced in the Q-nMT bundle only after phase I. It is worth noting, in this sense, that the percentage of cells with 2 colors Q-nMT bundle is analyzed only 1 hour after expression of mRuby-Tub1 was induced for phase I cells, but after 24 hours for phase II cells.We have modified Fig 1F and now provide images of cells after 3, 6 and 24h after glucose exhaustion and the corresponding percentage of cells displaying Q-nMT bundle with the two colors. We also now provide a western blot in Sup Fig 1H using specific antibodies against mTQZ (anti-GFP) and mRuby (anti-RFP).(6) In order to demonstrate that Q-nMT formation is an active process induced by a transient signal and that the Q-nMT bundle is required for cell survival, the authors treated cells with nocodazole for 24 h (Fig 1H and Supp Fig 1K). Both events, however, could be associated with the toxic effects of the extremely prolonged nocodazole treatment leading to cell death.

We have treated 5 days old cells for 24h with 30 µg/ml Noc. We then washed the drug and transferred the cells into a glucose free medium. We then followed both cell survival, using methylene blue, and the cell’s capacity to form a colony after refeeding. In these conditions, we did not observe any toxic effect of the nocodazole. This result is now provided in Sup Fig 1L and discussed line 172-176.

(7) The "Tub1-only" mutant displays shorter but stable Q-nMT bundles in phase II, although they are thinner than in wild-type cells. What happens in the "Tub3-only" mutant, which also has beta-tubulin levels similar to wild-type cells (Supp. Fig. 2B)?

In order to measure Q-nMT bundle length and thickness, we used Tub1 fused to GFP. This cannot be done in a Tub3-only mutant. Yet, we have measured Q-nMT bundle length in Tub3-only cells using Bim1-3GFP as a MT marker (as in Laporte et al, JCB 2013). As shown in the figure below, Q-nMT bundles were shorter in Tub3-only cells than in WT cells whatever the phase.

**Author response image 8. sa3fig8:** 

We do not know if this effect is directly linked to the absence of Tub1 or if it is very indirect and for example due to the fact that Tub1 and Tub3 interact differently with Bim1 or other proteins that are involved in Q-nMT bundle stabilization. As we cannot give a clear interpretation for that result, we decided not to present those data in our manuscript.

(8) Why were wild-type and ndc80-1 cells imaged after a 20 min nocodazole treatment to evaluate the role of KT-MT attachments in Q-nMT bundle formation (Fig 3A)? Importantly, this experiment is also missing a control in which Q-nMT length is analyzed in both wild-type and ndc80-1 cells at 25ºC instead of 37ºC.

In this experiment, we used nocodazole to test both the formation and the stability of the Q-nMT bundle. Fig 3A shows MT length distribution in WT (grey) and ndc80-1 (violet) cells expressing mTQZTub1 (green) and Nuf2-GFP (red), shifted to 37 °C at the onset of glucose exhaustion and kept at this non-permissive temperature for 12 or 96 h then treated with Noc. The control experiment was provided in Sup Fig 3B. Indeed, this figure shows MT length in WT (grey) and ndc80-1 (violet) expressing mTQZ-Tub1 (green) and Nuf2-GFP (red) grown for 4 d (96h) at 25 °C, and treated or not with Noc. This is now indicated in the text line 216 and in the figure legend line 976

**Author response image 9. sa3fig9:** 

(9) As a general comment linked to the previous concern, it is striking that in many instances, Q-nMT bundle length is measured after nocodazole treatment without any evident reason to do this and without displaying the results in untreated cells as a control. If nocodazole is used, the authors should explicitly indicate it and state the reason for it.

We provide control experiments without nocodazole for all of the figures. For the sake of figure clarity, for Fig.3A the control without the drug is in Sup. Fig. 3B, for Fig. 3B it is shown in Sup. Fig. 3D, for Fig. 4B, it is shown in Sup. Fig 4A. This is now stated in the text and in the figure legend: for Fig. 3A: line 216 and in the figure legend line 976; for Fig. 3B: line 222 and figure legend line 984; for Fig. 4B: line 280 and in the figure legend line 1017.

The only figures where the untreated cells are not shown is for Fig 1D since the goal of the experiment is to make dynamic MTs shorten.

In Fig. 5C and Sup. Fig. 5D to F, we used nocodazole to get rid of dynamic cytoplasmic MTs that form upon quiescence exit in order to facilitate Q-nMT bundle measurement. This was explained in our previous study (Laporte et al, JCB 2013). We now mention it in the figure legends, see for example Fig. 5 legend line 1054.

(10) Ipl1 inactivation using the ipl1-1 thermosensitive allele impedes Q-nMT bundle formation. The inhibitor-sensitive ipl1-as1 allele could have been further used to show whether this depends on its kinase activity, also avoiding the need to increase the temperature, which affects MT dynamics. As suggested, we have used the ipl1-5as allele. We have thus modified Fig 3B and now show that is it indeed the Ipl1 kinase activity that is required for Q-nMT bundle formation initiation (line 222).In any case, it is surprising that deletion of SLI15 does not affect Q-nMT formation (in fact, MT length is even larger), despite the fact that Sli15, which localizes and activates Ipl1, is present at the Q-nMT (Fig 3C). Likewise, deletion of BIR1 has barely any effect on MT length after 4 days in quiescence (Fig 3D). Do the previous observations mean that Ipl1 role is CPC-independent? Does the lack of Sli15 or Bir1 aggravate the defect in Q-nMT formation of ipl1-1 cells at non-permissive or semi-permissive temperature?

Thanks to the Reviewer’s comments, we have re-checked our sli15Δ strain and found that it was accumulating suppressors very rapidly. To circumvent this problem, we utilized the previously described sli15-3 strain (Kim et al, JCB 1999). We found that sli15-3 was synthetic lethal with both ipl1-1, ipl1-2 (as described in Kim et al, JCB 1999) and with ipl1-as5, preventing us from addressing the CPC dependence of the Ipl1 effect asked by the Reviewer. However, using the sli15-3 strain, we now show that inactivation of Sli15 upon glucose exhaustion does prevent Q-nMT bundle formation (See new Sup Fig 3F and the text line 226-227).

(11) Lack of both Bir1 and Bim1 act in a synergistic way with regard to the defect in Q-nMT bundle formation. Although the absence of both Sli15 and Bim1 is proposed to lead to a similar defect, this is not sustained by the data provided, particularly in the absence of nocodazole treatment (Supp. Fig 3E).

Deletion of bir1 alone has only a subtle effect on Q-nMT bundle length in the absence of Noc, yet in bir1Δ cells, Q-nMT bundles are sensitive to Noc. Deletion of BIM1 (bim1Δ) aggravates this phenotype (Fig. 3D). As mentioned above, Q-nMT bundle formation is impaired in sli15-3 cells. In our hands, and as expected from (Zimnaik et al, Cur Biol 2012), this allele is synthetic lethal with bim1Δ.

On the other hand, the simultaneous lack of Bir1 and Bim1 drastically reduces the viability of cells in quiescence and this is proposed to be evidence supporting that KT-MT attachments are critical for QnMT bundle assembly (Supp Fig 3G). However, similarly to what was indicated previously for the 24 h nocodazole treatment, here again, the lack of viability could be originated by other reasons that are associated with the lack of Bir1 and Bim1 and not necessarily with problems in Q-nMT formation. In fact, the viability defect of cells lacking Bir1 and Bim1 is similar to that of cells only lacking Bir1 (Supp Fig 3G).

We have previously shown that many mutants impaired for Q-nMT bundle formation (dyn1Δ, nip100Δ etc) have a reduced viability in quiescence (Laporte et al, JCB 2013). In the current study, a very strong phenotype is observed for other mutants impaired for Q-nMT bundle formation such as bim1Δ bir1Δ cells, but also for slk19Δ bim1Δ.

Importantly, as shown in the new Sup Fig 1L, in WT cells treated with Noc upon entry into quiescence, a treatment that prevents Q-nMT formation, showed a reduced viability, while a Noc treatment that does not affect Q-nMT bundle formation, i.e. a treatment in late quiescence, has no effect on cell survival. This solid set of data point to a clear correlation between the ability of cells to assemble a Q-nMT bundle and their ability to survive in quiescence. Yet, of course, we cannot formally exclude that in all these mutants, the reduction of cell viability in quiescence is due to another reason.

(12) Both Mam1 and Spo13 are, to my knowledge, meiosis-specific proteins. It is therefore surprising that mutants in these proteins have an effect on MT bundle formation (Fig 3G-H, Supp. Fig. 3G). Are Mam1 and Spo13 also expressed during quiescence? Transcription of MAM1 or SPO13 does not seem to be induced by glucose depletion in previously published microarray experiments, but if Mam1 are Spo13 are expressed in quiescent cells, the authors should show this together with their results.Indeed, it is interesting to notice that Mam1 and Spo13 are involved in both meiosis and Q-nMT bundle formation. As suggested by the Reviewer we have performed western blots in order to address the expression of those proteins in proliferation and quiescence (4d). We tagged Spo13 with either GFP, HA or Myc but none of the fusion proteins were functional. Yet, as shown in the new Sup Fig 3I, Mam1-GFP, Csm1-GFP and Lsr4-GFP were expressed both in proliferation and quiescence.(13) In the laser ablation experiments that demonstrate that KT-MT attachments are not needed in order to maintain Q-nMT bundles once formed, anaphase spindles of proliferating cells were cut as a control (Supp. Fig 3I). However, late anaphase cells have already segregated the chromosomes, which lie next to the SPBs (this can be evidenced by looking at Dad2-GFP localization in Supp. Fig 3I), so that only interpolar MTs are severed in these experiments. The authors should have instead used metaphase cells as a control, since chromosomes are maintained at the spindle midzone and the length and width of the metaphase spindle is more similar to that of the Q-nMT bundle.

We have tried to “cut” short metaphase spindles, but as they are < 1 µm, after the laser pulse, it is difficult to verify that spindles are indeed cut and not solely “bleached”. Furthermore, after the cut, the remaining MT structure that is detectable is very short, and we are not confident in our length measurements. Yet, this type of experiment has been done in *S. pombe* (Khodjakov et al, Cur Biol 2004 and Zareiesfandabadi et al, Biophys. J. 2022). In these articles the authors have demonstrated that after a cut, metaphase spindles are unstable and rapidly shrink through the action of Kinesin14 and dynein. This is now mentioned in the text line 265.

(14) In the experiment that shows that cycloheximide prevents Q-nMT disassembly after quiescence exit, and therefore that this process requires de novo protein synthesis (Fig. 5A), cells are indicated to express only Spc42-RFP and Nuf2-GFP. However, Stu2-GFP images are also shown next to the graph and, according to the figure legend, it was indeed Stu2-GFP that was used to measure individual QnMT bundles in cells treated with cycloheximide. In the graph, additionally, time t=0 represents the onset of MT bundle depolymerization, but Q-nMT bundle disassembly does not take place after cycloheximide treatment. The authors should clarify these aspects of the experiment.

Following the Reviewer’s suggestion, to clarify these aspects we have split Fig. 5A into 2 panels.

Finally, some minor issues are:(1) The text should be checked for proper spelling and grammar.

We have done our best.

(2) In some instances, there is no indication of how many cells were imaged and analyzed.

We now provide all these details either in the figure itself or in the figure legend.

(3) Besides the Q-nMT bundle, it is sometimes noticeable an additional strong cytoplasmic fluorescent signal in cells that express mTQZ-Tub1 and/or mRuby-Tub1 (e.g., Figs 1F, 1H and, particularly, Supp Fig 1H). What is the nature of these cytoplasmic MT structures?

We did mention this observation in the material and methods section (see line 526-528). This signal is a background fluorescence signal detected with our long pass GFP filter. It is not GFP as it is “yellowish” when we view it via the microscope oculars. This background signal can also be observed in quiescent WT cells that do not express any GFP. We do not know what molecule could be at the origin of that signal but it may be derivative of an adenylic metabolite that accumulates in quiescence and could be fluorescent in the 550nm –ish wavelength, but this is pure speculation.

(4) It is remarkable that a 20-30% decrease in tubulin levels had such a strong impact on the assembly of the Q-nMT bundle (Supp. Fig. 2). Can this phenotype be recovered by increasing the amount of tubulin in the mutants impaired for tubulin folding?

Yes, this is astonishing, but we believe our data are very solid since we observed that with both tub3Δ and in all the tubulin folding mutants we have tested (See Sup. Fig. 2). To answer Reviewer’s question, we would need to increase the amount of properly folded tubulin, in a tubulin folding mutant. One way to try to do that would be to find suppressors of GIM mutations, but this is a lengthy process that we feel would not add much strength to this conclusion.

(5) The graphs displaying the length of the Q-nMT bundle in several mutants in microtubule motors throughout a time course are presented in a different manner than in previous experiments, with data points for individual cells being only shown for the most extreme values (Fig 4C, 4H). It would be advisable, for the sake of comparison, to unify the way to represent the data.

We have now unified the way we present our figures.

(6) How was the exit from quiescence established in the experiments evaluating Q-nMT disassembly? How synchronous is quiescence exit in the whole population of cells once they are transferred to a rich medium?

We set the “zero” time upon cell refeeding with new medium. In fact, quiescence exit is NOT synchronous. We have reported this in previous publications, with the best description of this phenomena being in Laporte et al, MIC 2017.

The figures below are the same data but on the left graph, the kinetic is aligned upon SPB separation onset, while on the right graph (Fig 5A), it is aligned on MT shrinking onset.

**Author response image 10. sa3fig10:** 

We can add this piece of data in a Sup Figure if the Reviewer believes it is important.

**Reviewer #2 (Recommendations For The Authors):**
General:In general, more precise language that accurately describes the experiments would improve the text.We have tried to do our best to improve the text.The authors should clearly define what they mean by an active process and provide context to support this statement regarding the Q-nMT.

We have strived to clarify this point in the text (see paragraph form line 146 to 178).

It is reasonable to assume that structures composed of microtubules are dynamic during the assembly process. The authors should clarify what they mean by "stable by default i.e., intrinsically stable." Do they mean that when Q-nMT assembly starts, it will proceed to completion regardless of a change in condition?

We mean that in phase I the Q-nMT bundle is stabilized as it grows and that stabilization is concomitant with polymerization. By contrast, MTs polymerized during phase II are not stabilized upon elongation beyond the phase I polymer, and get stabilized later, in a separate phase (i.e. in phase III). We hope to have clarified this point in the text (see line 108-110).

In lines 33-34, the authors claim that the Q-nMT bundle functions as a "sort of checkpoint for cell cycle resumption." This wording is imprecise, and more significantly the authors do not provide evidence supporting a direct role for Q-nMT in a quiescence checkpoint that inhibits re-entry into the cell cycle.

We have softened and clarified the text in the abstract (see line 29-30)., in the introduction (line 101104), in the result section (line 331-332) and in the discussion (line 426-430).

Many statements are qualitative and subjective. Quantitative statements supported by the results should be used where possible, and if not possible restated or removed.

We provide statistical data analysis for all the figures.

The number of hours after glucose exhaustion used for each phase varies between assays. This is likely a logistical issue but should be explained.

This is indeed a logistical issue and when pertinent, it is explained in the text.

It would be interesting to address how this process occurs in diploids. Do they form a Q-nMT? How does this relate to the decision to enter meiosis?

Diploid cells enter meiosis when they are starved for nitrogen. Upon glucose exhaustion diploids do form a Q-nMT bundle. This is shown and measured in the new Sup Fig1C. In fact, in diploids, Q-nMT bundles are thicker than in haploid cells.

It would be interesting to address how the timescale of this process compares to the types of nutrient stress yeast would be exposed to in the environment.

We have transferred proliferating yeast cells to water, to try to mimic what could happen when yeast cells face rain in the wild. As shown below, they do form a Q-nMT bundle that becomes nocodazole resistant after 30h. This data is now provided in the new Sup Fig 1D.

It is recommended that the authors use FRAP experiments to directly measure the stability of the QnMT bundles.

This experiment was published in (Laporte et al, 2013). Please see response to Reviewer #1.

In many cases, the description of the experimental methods lacks sufficient detail to evaluate the approach or for independent verification of results.

We have strived to provide a more detailed material and methods section, as well as more detailed figure legends and statistical informations.

Specific comments on figures:In Figure 1 c, what do the polygons represent? They do not contain all the points of the associated colour.

The polygon represented the area of distribution of 90% of the data points. As they did not significantly add to the data presentation they have been removed.

In Figure 2 a, is the use of two different sets of markers to control for the effect of the markers on microtubule dynamics?

Yes, we are always concerned about the influence of GFP on our results, so very often we replicate our experiments with different fluorescent proteins or even with different proteins tagged with GFP. This is now mentioned in the text (line 184-186).

Is it accurate to say (line 201, figure 3 a) that no Q-nMT bundles were detected in ndc80-1 cells shifted to 37 degrees, or are they just shorter?

As shown in Fig 3A, in ndc80-1 cells, most of the MT structures that we measured are below 0,5um. This has been re-phrased in the text (line 214-215).

Lines 265-269, figure 4 b, how can the phenotype observed in cin8∆ cells be explained given the low abundance of Cin8 that is detected in quiescent cells?

Faint fluorescence signal is not synonymous of an absence of function. As shown in Sup Fig 4B, we do detect Cin8-GFP in quiescent cells.

Quantification is needed in Figure 4 panels c and h.

Fig 4C and 4H have been changed and quantification are provided in the figure legend.

**Reviewer #3 (Recommendations For The Authors):**
A few points should be addressed for clarity:(1) Sup. Fig. 1K: are only viable cells used for the colony-forming assay? How were these selected? If not, the assay would just measure survival (as in the viability assay).

Yes, only viable cells were selected for the colony forming assay. We used methylene blue to stain dead cells. Then, we used a micromanipulation instrument (Singer Spore Play) that is commonly used for tetrad dissection to select “non blue cells” and position them on a plate (as we do with spores). Each micromanipulated cell is then allowed to grow on the plate and we count colonies (see picture in Sup Fig 1L right panel). This was described in Laporte et al, JCB 2011. We have added that piece of information in the legend (line 1129-1130) and in the M&M section (line 580-586).

(2) Could Tub3 have a role in phase I? It is not clear why the authors conclude involvement only in phase II.

As it can be seen in Fig 2D, MT bundle length and thickness are quite similar in WT and Tub1-only cells in phase I, indicating that the absence of Tub3 as no effect in phase I. In Tub1-only cells, MT bundles are thinner in both phase II and phase III, yet, they get fully stabilized in phase III. Thus, the effect of Tub3 is largely specific to the nucleation/elongation of phase II MTs. We hope to have clarified that point in the text (line 203-207).

(3) Quantifications, statistics: for all quantifications, the authors should clearly state the number of experiments (replicates), and number of cells used in each, and what number was used for statistics. For all quantifications in cells, it seems that the values from the total number of cells across different experiments were plotted and used for statistics. This is not very useful and results in extremely small p values. I assume that the values for individual cells were obtained from multiple, independent experiments. Unless there are technical limitations that allow only a very small sample size (not the case here for most experiments), for experiments involving treatments the authors should determine values for each experiment and show statistics for comparison between experiments rather than individual cells pooled from multiple experiments.

All the experiments have been done at least in replicate. In the new Fig. 1A, we now display each independent experiment with a specific color code. For Fig 2B and 2C we now provide the data obtained for each separate experiment in Sup Fig 2C. Additional details about quantifications and statistics are provided in the M&M section or in the specific figure legends.